# Telomerase RNA component knockout exacerbates *Staphylococcus aureus* pneumonia by extensive inflammation and dysfunction of T cells

Yasmina Reisser[1]*, Franziska Hornung[1], Antje Häder[1], Thurid Lauf[1,2], Sandor Nietzsche[3], Bettina Löffler[1], Stefanie Deinhardt-Emmer[1]*

[1]Institute of Medical Microbiology, Jena University Hospital, Jena, Germany; [2]Else Kröner Graduate School for Medical Students 'JSAM', Jena University Hospital, Jena, Germany; [3]Center for Electron Microscopy, Jena University Hospital, Jena, Germany

## eLife Assessment

In this manuscript, the authors sought to elucidate mechanistic intricacies of inflammatory responses, with emphasis on T cell dysfunction, to *S. aureus*-induced pneumonia in the context of aging process using Terc deficient mice. Conceptually, the study is very interesting with a set of **useful** findings. Although some experimental approaches are appropriate, the work as shown in the revised manuscript remains significantly underpowered and the absence of rigorous controls make this study **incomplete** in support of its claims.

**\*For correspondence:**
yasmina.reisser@med.uni-jena.de (YR);
stefanie.deinhardt-emmer@med.uni-jena.de (SDE)

**Competing interest:** The authors declare that no competing interests exist.

**Abstract** The telomerase RNA component (Terc) constitutes a non-coding RNA critical for telomerase function, commonly associated with aging and pivotal in immunomodulation during inflammation. Our study unveils heightened susceptibility to pneumonia caused by *Staphylococcus aureus* (*S. aureus*) in *Terc* knockout (*Terc*$^{ko/ko}$) mice compared to both young and old infected counterparts. The exacerbated infection in *Terc*$^{ko/ko}$ mice correlates with heightened inflammation, manifested by elevated interleukin-1β (IL-1β) levels and activation of the NLR family pyrin domain containing 3 (NLRP3) inflammasome within the lung. Employing mRNA sequencing methods alongside in vitro analysis of alveolar macrophages (AMs) and T cells, our study elucidates a compelling correlation between *Terc*$^{ko/ko}$, inflammation, and impaired T cell functionality. *Terc* deletion results in compromised T cell function, characterized by dysregulation of the T cell receptor and absence of CD247, potentially compromising the host's capacity to mount an effective immune response against *S. aureus*. This investigation provides insights into the intricate mechanisms governing increased vulnerability to severe pneumonia in the context of Terc deficiency, which might also contribute to aging-related pathologies, while also highlighting the influence of Terc on T cell function.

## Introduction

As the global population ages, the impact of infectious diseases on the aging population becomes increasingly significant. Among the most relevant risk factors leading to higher susceptibility as well as severity in both community-acquired (CA) and healthcare-associated (HA) pneumonia is advanced age (≥60 years) (*Torres et al., 2021*). Particularly severe is pneumonia caused by *Staphylococcus aureus* (*S. aureus*) (*Lee et al., 2022*). *S. aureus* is a gram-positive bacterium that frequently colonizes humans but can manifest various pathologies, ranging from mild to more severe infections, including pneumonia

(*Howden et al., 2023*; *Jong et al., 2019*). It accounts for a notable proportion of CA and HA cases (*Lee et al., 2022*; *Torres et al., 2021*). The bacterium exhibits a broad spectrum of virulence factors, including toxins, adhesins, and immune evasion strategies, contributing to its ability to establish infections in the lower respiratory tract (*Howden et al., 2023*). Aging is a central factor predisposing individuals to a higher risk for *S. aureus* pneumonia (*Torres et al., 2021*), as aging is impacting several important functions of the body such as the immune response and thus the capability to mount an effective defensive response (*Montecino-Rodriguez et al., 2013*). Aging-related dysregulation of the immune response is also characterized by inflammaging, defined as the presence of elevated levels of pro-inflammatory cytokines in the absence of an obvious inflammatory trigger (*Franceschi et al., 2000*; *Mogilenko et al., 2022*). Additionally, immune cells, such as macrophages, exhibit an activated state that alters their response to infection (*Canan et al., 2014*). In contrast, the immune response of macrophages to infectious challenges has been shown to be initially impaired in aged mice (*Boe et al., 2017*). Thus aging is a relevant factor impacting the pulmonary immune response. This highlights the vital connection between aging and *S. aureus* pneumonia, as well as the need for further research investigating this connection.

The process of aging is a multifaceted phenomenon that involves various underlying mechanisms, known as hallmarks (*López-Otín et al., 2023*). Among these hallmarks, telomere shortening is a significant factor in cellular aging. Telomeres serve as protective caps at the end of chromosomes, which shorten with every cell division until they reach a critical length where replication becomes impossible, eventually resulting in either apoptosis or senescence (*López-Otín et al., 2023*). However, this process can be counteracted by the telomerase, which can add new repeats to the ends of chromosomes and thus prevent the shortening of telomeres (*Greider and Blackburn, 1985*). The telomerase mainly consists of two parts: the telomerase reverse transcriptase (Tert), which harbors the enzymatic activity, and Terc, which serves as a template for the addition of new repeats (*Sahin and Depinho, 2010*; *Shay and Wright, 2019*). Telomerase activity is restricted to specific tissues and cell types, largely dependent on the expression of *Tert*. While *Tert* is highly expressed in stem cells, progenitor cells, and germline cells, its expression is minimal in most differentiated cells (*Chakravarti et al., 2021*). Consequently, the impact of telomerase dysfunction on tissues varies according to their self-renewal rate (*Chakravarti et al., 2021*).

One important aspect of telomere dysfunction is the impact of telomere shortening on the immune system as well as the hematopoietic system. Tissues or organ systems that are highly replicative, such as the skin or the hematopoietic system, are affected first by telomere shortening (*Chakravarti et al., 2021*). Due to the loss of the protective telomere caps, mutations can accumulate, and the cells ultimately become senescent (*Chakravarti et al., 2021*). Interestingly, while most differentiated cells do not exhibit telomerase activity, T cells display a high activity of this enzyme (*Hodes et al., 2002*). Aging T cells become senescent, which is characterized by short telomeres and the absence of CD28 expression (*Hohensinner et al., 2011*). Furthermore, telomere length and T cell functionality are closely connected. T cells with shorter telomeres are implicated in diseases related to aging like arthritis (*Hohensinner et al., 2011*). Additionally, CD4$^+$ T cells from *Terc* knockout (*Terc*$^{ko/ko}$) mice displayed several characteristics of aged T cells such as a reduction in CD28 expression and changed secretion of cytokines. Furthermore the naïve T cell population was found to be reduced (*Matthe et al., 2022*). This highlights the importance of intact telomeres and telomerase on immune cell function.

While most differentiated cells lack the expression of *Tert*, almost all human cells and several murine tissues express *Terc* (*Zhang et al., 2018*). However, the function of ubiquitous *Terc* expression remains elusive (*Chakravarti et al., 2021*; *Shay and Wright, 2019*). Interestingly, recent studies identified a variety of functions of Terc, which are independent of its function as part of the telomerase. Notably, a study could show that Terc triggers inflammation via the nuclear factor kappa-light-chain-enhancer of activated B cells (NF-κB) pathway (*Liu et al., 2019*). Furthermore, Terc was identified to be involved in cell proliferation by enhancing the expression of several genes belonging to the phosphoinositide 3-kinase (Pi3K)-protein kinase B (AKT) pathway (*Wu et al., 2022*). The same study could show that this mechanism also contributed to CD4$^+$ T cell expansion (*Wu et al., 2022*). Additionally, *Terc* deletion in mice induced NLR family pyrin domain containing 3 (NLRP3) inflammasome activation in macrophages upon pulmonary infection with *S. aureus*, thereby revealing another connection between immune modulation and the presence of Terc (*Kang et al., 2018*). Thus, it appears that Terc could serve as a

relevant immunomodulatory factor, potentially implicated in *S. aureus* - induced pathologies. Interestingly, downregulation of *Terc* and *Tert* expression in tissues of aged mice and rats has been found (*Tarry-Adkins et al., 2021*; *Zhang et al., 2018*). Therefore, as a potential immunomodulatory factor reduced *Terc* expression could be connected to aging-related pathologies.

Knockout of *Terc* in mice is a well-established model for premature aging, as telomeres of *Terc*[ko/ko] mice shorten with every generation (*Wong et al., 2009*). Additionally, the telomeres shorten during each generation with increasing age of the mice (*Wong et al., 2009*). Furthermore, third-generation (G3) *Terc*[ko/ko] mice with a C56Bl/6 background have a shorter lifespan, a smaller litter size, and a generally reduced body size and weight (*Wong et al., 2009*). Although G3 *Terc*[ko/ko] mice with shortened telomeres were used in this study, they were infected at a young age (8 weeks). This approach allowed for the investigation of *Terc* deletion effects rather than telomere dysfunction. As control cohort age-matched young WT mice were utilized. To investigate whether *Terc* deletion, beyond critical telomere shortening, impacts the pulmonary immune response, we used young *Terc*[ko/ko] mice. Additionally, naturally aged mice (2 years of age) were infected to explore the potential link to a fully developed aging phenotype.

In this study, we aimed to investigate the impact of *Terc* deletion, as an essential immunomodulatory factor, on the immune response to *S. aureus* - induced pneumonia in mice. For this purpose, G3 *Terc*[ko/ko] mice were infected with *S. aureus*, and the progression of disease and immune response were evaluated. Strikingly, our data highlights that the deletion of *Terc* resulted in a more severe disease outcome and disrupted the innate and adaptive immune response. These findings suggest a possible connection between Terc and immune cell homeostasis.

## Results

### *Terc* knockout is associated with severe pneumonia in mice

To investigate the impact of Terc on the course of *S. aureus* pneumonia, we compared young *Terc*[ko/ko] mice to young and old wild-type (WT) mice. For this purpose, we performed an intranasal infection of *Terc*[ko/ko] mice at the age of 8 weeks and naturally aged mice at the age of 2 years (old WT) with the *S. aureus* strain USA300. WT mice at the age of 8 weeks (young WT) were additionally infected and utilized as the control group (*Figure 1A*).

The clinical score of infected *Terc*[ko/ko] mice was increased compared to the other two infected mice cohorts at 24 hr post infection (hpi) (*Figure 1A*). The bacterial load was significantly higher in the bronchoalveolar lavage (BAL) of *Terc*[ko/ko] mice compared to old WT mice (*Figure 1B*). Interestingly, three distinct degrees of severity were observed in the infected *Terc*[ko/ko] mice (n=5): mild infection, systemic infections, or fatal infections (*Figure 1—figure supplement 1A*). Mice exhibiting systemic infection were characterized by the presence of bacteria in extrapulmonary organs, specifically observed in the liver and kidney tissues of some *Terc*[ko/ko] mice (*Figure 1—figure supplement 1B*).

Infection of young WT mice resulted in a significant decrease in body weight at 24 hpi. The body weight of infected *Terc*[ko/ko] mice was reduced compared to the infected old WT mice and the non-infected *Terc*[ko/ko] mice (*Figure 1C*). In contrast, at 24 hpi the relative weight of the lung was increased in infected compared to the non-infected mice for all cohorts, indicating inflammation of the lung (*Figure 1D*). However, this effect was the most pronounced in young WT mice.

Based on our observation of the different degrees of severity displayed by the infected *Terc*[ko/ko] mice, we categorized them according to their clinical score and the presence of bacteria in extrapulmonary organs, distinguishing between mice with and without systemic infection. This grouping resulted in more pronounced differences between the mice cohorts regarding bacterial load and reduction of relative body weight (*Figure 1—figure supplement 1C and D*). The relative lung weight increased for all groups compared to the respective non-infected mice, except for the *Terc*[ko/ko] mouse without systemic infection (*Figure 1—figure supplement 1E*). Additionally, average telomere length of the lungs of young and old WT as well as *Terc*[ko/ko] mice was measured. As shortened telomeres are a hallmark of aging, impacting other cellular mechanisms such as cellular senescence, they could have an influence on the disease progression (*López-Otín et al., 2023*). Notably, the telomeres of *Terc*[ko/ko] mice were the shortest among the three different mouse models, even shorter than those of naturally aged mice (*Figure 1—figure supplement 1F*).

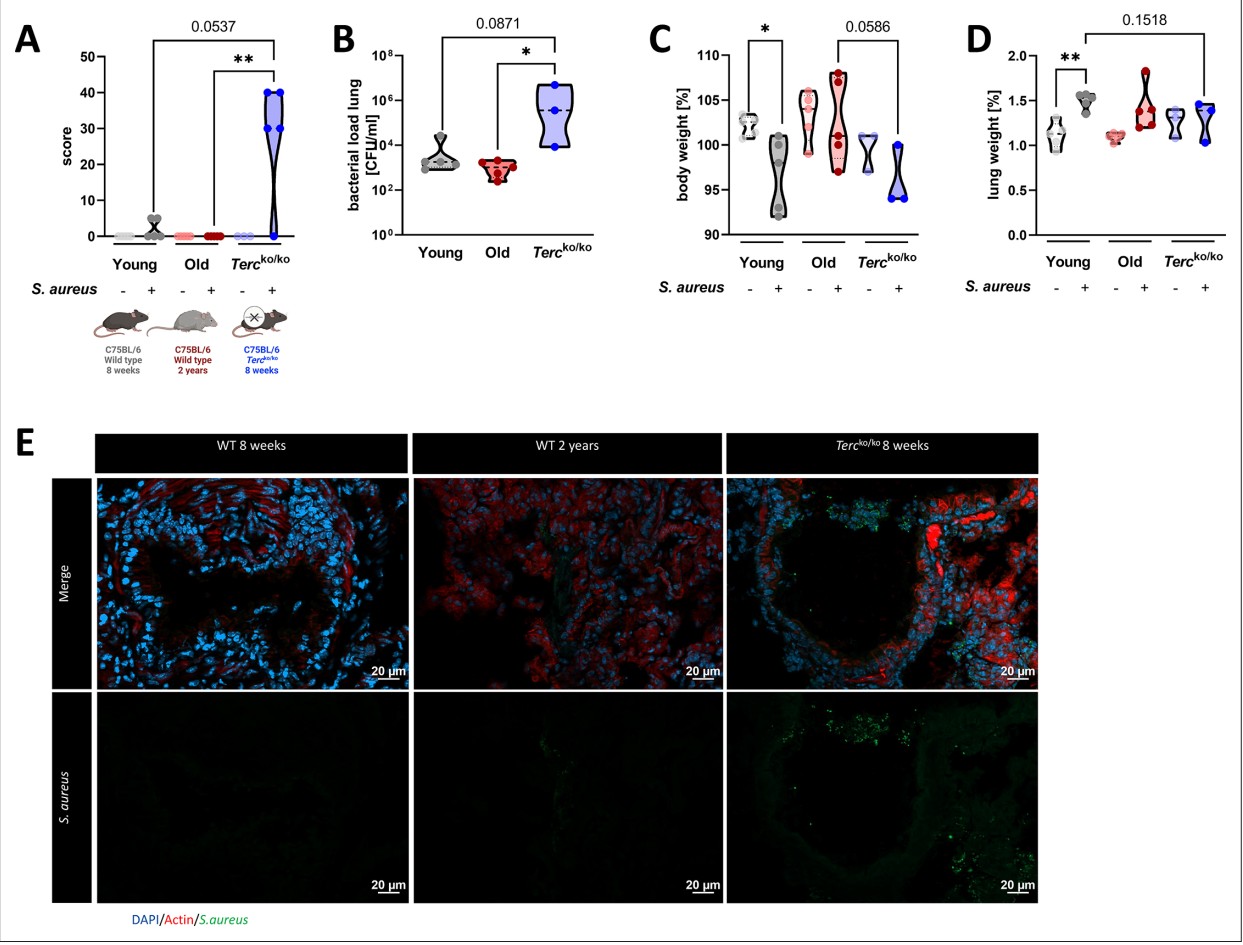

**Figure 1.** $Terc^{ko/ko}$ mice exhibit more severe pneumonia with increased mortality and a higher bacterial load. (**A**) Clinical score of non-infected and infected young wild-type (WT) (n=10, 5 non-infected and 5 infected), old WT (n=10, 5 non-infected and 5 infected), and $Terc^{ko/ko}$ (n=8, 3 non-infected and 5 infected) mice at 24 hr post infection (hpi). The age and genetic background of the different groups is depicted below. (**B**) Bacterial load of the lungs of infected mice at 24 hpi. Data is displayed as logarithmic. (**C**) Relative body weight of young WT, old WT, and $Terc^{ko/ko}$ mice. Relative body weight displays body weight at 24 hpi as a percentage of the body weight at the time of infection. (**D**) Relative lung weight of young WT, old WT, and $Terc^{ko/ko}$ mice. Relative lung weight displays lung weight at 24 hpi as a percentage of the current body weight. (**E**) Immunofluorescence staining of lung tissue of infected young (WT 8 weeks), old (WT 2 years), and $Terc^{ko/ko}$ mice. Lungs were stained for *S. aureus* (green), actin (red), and DAPI (blue). Representative pictures are shown for each group. *p<0.05, **p<0.01, ***p<0.001. p-Values calculated by Kruskal-Wallis test (**A–D**). Data is displayed as violin plot showing the median as well as lower and upper percentile of each dataset. Each replicate is displayed as a data point. Created with BioRender.com.

The online version of this article includes the following source data and figure supplement(s) for figure 1:

**Source data 1.** Numeric mice data for *Figure 1A–D* and *Figure 1—figure supplement 1A–E*.

**Figure supplement 1.** $Terc^{ko/ko}$ mice with systemic infection display more severe pneumonia with bacteremia and increased mortality.

**Figure supplement 1—source data 1.** Numeric data for *Figure 1—figure supplement 1F*.

**Figure supplement 1—source data 2.** Numeric data for *Figure 1—figure supplement 1H*.

In order to investigate gene expression patterns, mRNA sequencing of the lungs was performed. This data supported the grouping of infected $Terc^{ko/ko}$ mice into different groups, as the principal component analysis plot showed a separate cluster containing $Terc^{ko/ko}$ mice with systemic infection (*Figure 1—figure supplement 1G*).

Immunofluorescence (IF) staining of *S. aureus* within the lung tissue of all three infected mice cohorts was carried out. The increased signal for *S. aureus* in sections of $Terc^{ko/ko}$ mice compared to sections of young WT and old WT mice closely reflects the quantified colony-forming units (CFU) detected in the BAL (*Figure 1E*).

$Terc^{ko/ko}$ mice were characterized by an increased bacterial load in the lungs, a higher clinical score, as well as mortality (*Figure 1A and B*; *Figure 1—figure supplement 1A*). Additionally, we could

observe different degrees of severity in the infected *Terc*<sup>ko/ko</sup> cohort and could identify a subgroup of mice with systemic infection (*Figure 1—figure supplement 1A, B, and G*).

## *Terc*<sup>ko/ko</sup> mice showed excessive inflammation with inflammasome activation

In order to investigate the underlying pathomechanisms leading to a more severe course of disease in the *Terc*<sup>ko/ko</sup> mice, inflammation parameters in the lungs were measured. In the course of this analysis, we applied the previously outlined groups of mice, differentiating between those with and without systemic infection in the *Terc*<sup>ko/ko</sup> cohort. Several pro-inflammatory cytokines were significantly upregulated in *Terc*<sup>ko/ko</sup> mice with systemic infection, such as interleukin-6 (IL-6), granulocyte-macrophage colony-stimulating factor, and interleukin-1β (IL-1β). Other important pro-inflammatory factors like tumor necrosis factor α (TNFα) and interleukin-1α (IL-1α) were also elevated (*Figure 2A*). A significant upregulation of pro-inflammatory mediators due to infection was, however, solely detected in infected *Terc*<sup>ko/ko</sup> mice compared to the corresponding non-infected control cohort (*Figure 2A*).

We performed mRNA sequencing of the murine lung tissue of infected and non-infected mice at 24 hpi to elucidate potential differentially expressed genes (DEGs) that contribute to the more severe illness of *Terc*<sup>ko/ko</sup> mice. Comparison of overall gene expression pattern of the different groups revealed that non-infected young WT and *Terc*<sup>ko/ko</sup> mice cluster together and show a similar expression pattern. In contrast, the infected mice of those cohorts display different expression patterns (*Figure 2B*). This highlights that most differences between the knockout and WT mice are only present after infection with *S. aureus*. Furthermore, both non-infected and infected old WT mice show a completely distinct expression profile compared to young WT as well as *Terc*<sup>ko/ko</sup> mice (*Figure 2B*).

Intriguingly, several genes related to the NLRP3 inflammasome pathway, such as *NLRP3* or thioredoxin interacting protein (*TXNIP*), were upregulated in infected *Terc*<sup>ko/ko</sup>, indicating activation of the inflammasome (*Figure 2C*; *Figure 2—figure supplement 1A*). Notably, expression of genes encoding for TNFα, IL-1β, and IL-6 were heightened. Thus, the increased inflammation in the lungs of *Terc*<sup>ko/ko</sup> mice is likely induced by activation of the NLRP3 inflammasome (*Figure 2C*; *Figure 2—figure supplement 1A*). Supporting this, the nucleotide oligomerization domain-like receptor signaling pathway was among the most significantly enriched pathways when comparing infected *Terc*<sup>ko/ko</sup> and young WT mice (*Figure 2D*).

Macrophages are part of the first line of defense against bacterial infections as they can phagocytose bacteria and support pathogen clearance. Alveolar macrophages (AMs) are present in the lung and can directly react to the pathogen at the site of infection via the production of inflammatory mediators or attraction of other immune cells (*Malainou et al., 2023*). To specifically investigate the role of AMs during *S. aureus* infection, we isolated them from lungs of young WT and *Terc*<sup>ko/ko</sup> mice. By using scanning electron microscopy (SEM) we could strikingly visualize the ability of AMs to phagocytose *S. aureus* (*Figure 3A*). Additionally, we identified the AMs via IF staining of the macrophage marker CD68 (*Figure 2—figure supplement 1B*).

Notably, our sequencing data revealed an upregulation of pro-inflammatory M1 macrophage markers, such as C-X-C motif chemokine ligand 2(*CXCL2*), in *Terc*<sup>ko/ko</sup> mice following infection (*Figure 3B*; *Figure 2—figure supplement 1C*). Anti-inflammatory M2 markers, e.g., interferon regulatory factor 4 (*IRF4*), were mainly downregulated. Additionally, cluster of differentiation 14 (*CD14*), a general macrophage marker, was increased in the infected *Terc*<sup>ko/ko</sup> mice (*Figure 3B*; *Figure 2—figure supplement 1C*). Furthermore, the expression of multiple chemokines such as C-X-C motif ligand 1–3 (*CXCL1-3*) and C-C motif chemokine ligand 2 (*CCL2*) was elevated, suggesting increased migration of leukocytes such as macrophages and neutrophils to the lung (*Figure 3C*; *Figure 2—figure supplement 1D*). Additionally, the TNF, NF-κB, and IL-17 signaling pathways were among the most significantly enriched pathways in infected *Terc*<sup>ko/ko</sup> mice compared to young WT mice (*Figure 2D*).

Combining the findings above, *S. aureus* infection in *Terc*<sup>ko/ko</sup> mice leads to excessive inflammation induced by NLRP3 inflammasome activation and infiltration of pro-inflammatory immune cells, such as M1 macrophages.

## The T cell receptor is partially downregulated in infected *Terc*<sup>ko/ko</sup> mice

In addition to macrophages, T cells and their activation play a crucial role in the immune response and are vital for resolving infections. Surprisingly, multiple general T cell markers such as *CD2*, *CD5*,

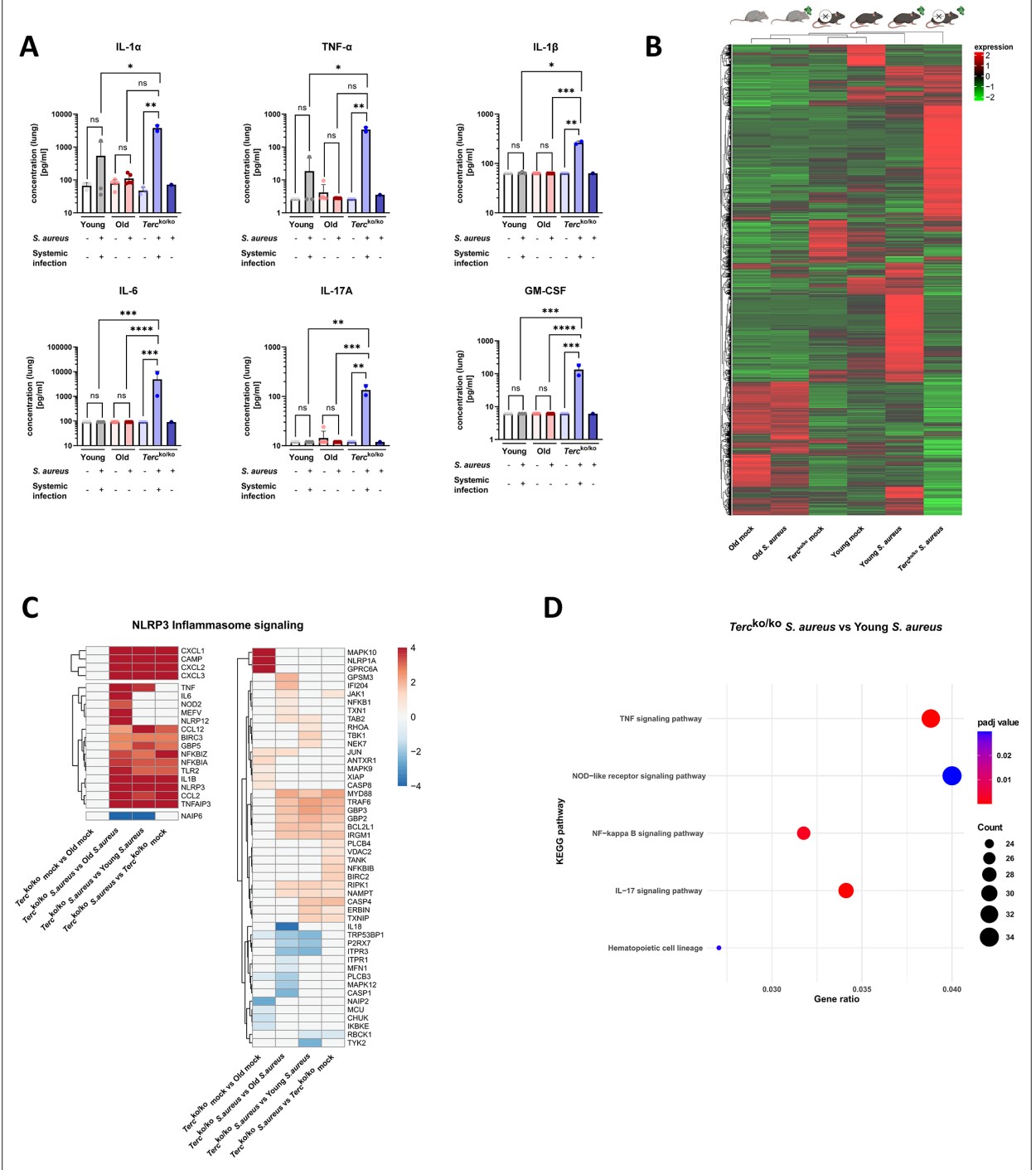

**Figure 2.** Lungs of *Terc*^ko/ko^ mice display excessive inflammation and NLRP3 inflammasome activation. (**A**) Pro-inflammatory cytokines measured in lung homogenates of non-infected and infected young wild-type (WT) (each n=3), old WT (each n=5), and *Terc*^ko/ko^ mice (non-infected: n=3, with systemic infection: n=2, and without systemic infection: n=1). Infected *Terc*^ko/ko^ mice were grouped into mice with systemic and without systemic infection. Cytokines were measured using a flow cytometry-based LEGENDPlex mouse inflammation panel. Data is displayed on a logarithmic scale. (**B**) Heatmap of overall gene expression in the lungs of the different mice cohorts at 24 hr post infection (hpi). Gene expression is displayed as the log2 of the fragments per kilobase of transcript per million fragments mapped (fpkm) of each individual gene. Gene expression is normalized to sequencing depth and transcript length. Red indicates upregulation, and green indicates downregulation of the gene expression. The respective mouse groups are visualized above the heatmap. For each mouse cohort three biological replicates were sequenced. (**C**) Heatmap of differentially expressed genes (DEGs) belonging to the NLRP3 inflammasome pathway. Red indicates upregulation, and blue indicates downregulation of the gene. Differential gene expression analysis was conducted using the DESeq2 software as well as negative binomial distribution. For false discovery rate (FDR) calculation the

*Figure 2 continued on next page*

*Figure 2 continued*

Benjamini-Hochberg procedure was used. Heatmaps were constructed using RStudio. Only significant DEGs (p-value<0.05) were displayed in the heatmap. Non-significant genes were set to zero and are shown as white. DEGs are displayed as log2fold-change. For each group three biological replicates were sequenced and used for analysis. For clarity, the heatmap has been split in half, and no column clustering has been performed. The respective groups used for the comparison are indicated below each column. (D) KEGG pathway enrichment analysis comparing infected young and *Terc*ko/ko mice at 24 hpi. The gene ratio describes the ratio of differentially expressed genes to all genes of the respective pathway. The dot plot was constructed using RStudio. All sequencing data displays mRNA sequencing data of non-infected and infected lungs of the three different mice cohorts at 24 hpi. ns≥0.05, *p<0.05, **p<0.01, ***p<0.001, ****p<0.0001. p-Values calculated by Kruskal-Wallis test (A). Data is presented as mean ± SD. Each replicate is displayed as a data point.

The online version of this article includes the following source data, source code, and figure supplement(s) for figure 2:

**Source code 1.** R code for heatmap construction.

**Source code 2.** R code for KEGG enrichment analysis dot plot construction.

**Source data 1.** Numeric data for *Figure 2A*.

**Source data 2.** Numeric data for *Figure 2B*.

**Source data 3.** Numeric data for *Figure 2C*.

**Source data 4.** Numeric data for *Figure 2D*.

**Figure supplement 1.** Several pathways relevant for inflammation and adaptive and innate immune response are differentially expressed in infected *Terc*ko/ko mice.

**Figure supplement 1—source data 1.** Numeric data for *Figure 2—figure supplement 1A*.

**Figure supplement 1—source data 2.** Numeric data for *Figure 2—figure supplement 1C*.

**Figure supplement 1—source data 3.** Numeric data for *Figure 2—figure supplement 1D*.

**Figure supplement 1—source data 4.** Numeric data for *Figure 2—figure supplement 1E*.

**Figure supplement 1—source data 5.** Numeric data for *Figure 2—figure supplement 1F*.

and *CD4* were downregulated in the infected *Terc*ko/ko mice compared to young WT mice (*Figure 4A*; *Figure 2—figure supplement 1E*). However, these distinctions are not evident in comparisons between non-infected *Terc*ko/ko and young WT mice. This further emphasizes that the observed variations in immune response activation arise specifically in response to infection (*Figure 4A*). Some T cell markers, such as *CD5*, that were downregulated in infected *Terc*ko/ko mice were also downregulated in infected old WT mice compared to young WT mice (*Figure 2—figure supplement 1E*).

To rule out that this effect is caused by a reduction of T cells in *Terc*ko/ko mice, the immune cells of non-infected *Terc*ko/ko, young WT and old WT mice were analyzed via flow cytometry. However, no significant difference was observed between the groups (*Figure 4B*).

Additionally, several genes belonging to the T cell receptor (TCR) signaling pathway were down-regulated in infected *Terc*ko/ko mice, such as *CD3G* (encoding CD3γ) and *CD247* encoding CD3 ζ (*Figure 4C*, *Figure 2—figure supplement 1F*). Expression of downstream factors of the TCR like lymphocyte cell-specific protein-tyrosine kinase (*LCK*), linker of activation (*LAT*), and IL-2 inducible T cell kinase (*ITK*) was also reduced (*Figure 4C*, *Figure 2—figure supplement 1F*). We also noted several of these distinctions, including the downregulation of *LCK* and *ITK* when comparing infected old WT and young WT mice (*Figure 4C*). Since there was no reduction of T cells in *Terc*ko/ko mice, the change in TCR signaling and marker expression is likely induced by the infection.

Specifically, *CD247* was the only gene downregulated compared to infected young WT, old WT, and non-infected *Terc*ko/ko mice (*Figure 4C*). Additionally, *CD247* was downregulated in infected old WT mice compared to infected young WT mice (*Figure 4C*). CD247 is an essential factor for the assembly of the TCR and the downstream signaling after stimulation with an antigen (*Jin et al., 2024*; *Pitcher et al., 2003*). In young WT mice gene expression levels of *CD247* were increased with infection, likely due to proliferation and infiltration of T cells (*Figure 4D*). Notably, in infected *Terc*ko/ko mice, the expression of *CD247* is absent. Moreover, there is no difference in *CD247* expression between non-infected and infected old WT mice (*Figure 4D*). Similar trends can be seen for other T cell markers, such as programmed cell death 1 (*PDCD1*) and *CD4* (*Figure 4E and F*). Additionally, the TCR signaling pathway and Th1 and Th2 differentiation pathway are among the most enriched pathways in infected young WT mice compared to non-infected young WT mice. In contrast, those pathways are

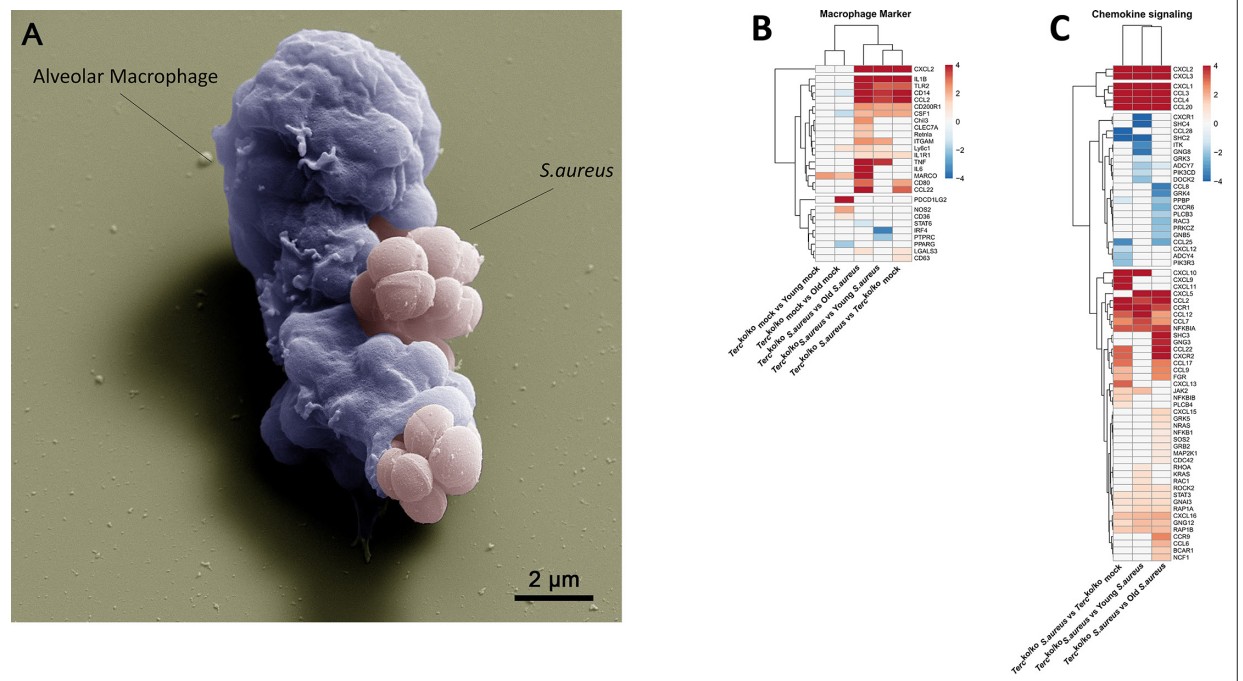

**Figure 3.** Elevated expression of pro-inflammatory M1 macrophages markers and chemokines are present in the lungs of infected *Terc*ko/ko mice. (**A**) Colorized scanning electron microscopy (SEM) picture of an alveolar macrophage (AM) isolated from young wild-type (WT) mice phagocytosing *S. aureus*. Heatmap of differentially expressed macrophage markers (**B**) and differentially expressed genes (DEGs) belonging to the chemokine signaling pathway (**C**). Red indicates upregulation, and blue indicates downregulation of the gene. Differential gene expression analysis was conducted using the DESeq2 software as well as negative binomial distribution. For false discovery rate (FDR) calculation the Benjamini-Hochberg procedure was used. Heatmaps were constructed using RStudio. Only significant DEGs (p-value<0.05) were displayed in the heatmap. Non-significant genes were set to zero and are shown as white. DEGs are displayed as log2fold-change. For each group three biological replicates were sequenced and used for analysis. The respective groups used for the comparison are indicated below each column. All heatmaps display mRNA sequencing data of non-infected and infected lungs of the three different mice cohorts at 24 hr post infection (hpi).

The online version of this article includes the following source data for figure 3:

**Source data 1.** Numeric data for *Figure 3B*.

**Source data 2.** Numeric data for *Figure 3C*.

not upregulated when comparing infected *Terc*ko/ko with non-infected *Terc*ko/ko mice (*Figure 5—figure supplement 1A*).

These findings indicate a malfunction in the activation of T cells as components of the TCR signaling pathway were partially downregulated. The complete absence of *CD247*, an essential factor for downstream activation of the TCR, indicates aberrant T cell function of *Terc*ko/ko mice during infection.

## T cells of *Terc*ko/ko mice do not adequately react to infectious challenge

To investigate the functionality of T cells and their CD247 expression in *Terc*ko/ko mice during infection, murine T cells and AMs were isolated. An overactive inflammatory response could be a potential explanation for the dysregulated TCR signaling. Therefore, to study the impact of AMs on T cells we infected the co-culture of both isolated cells with a multiplicity of infection 5 (MOI5) of heat-killed (HK) *S. aureus* for 24 hr (*Figure 5A*). All components of the experimental setup were visualized using SEM. In particular, the close association between AMs and T cells was notable (*Figure 5B*). IF staining of AMs of *Terc*ko/ko and young WT mice revealed the successful internalization of *S. aureus* as well as a change of phenotypic appearance to a more activated shape (*Figure 5C*).

Utilizing flow cytometry, we observed a substantial elevation in the baseline expression of CD247 in T cells from *Terc*ko/ko mice in comparison to T cells from young WT mice at 24 hpi (*Figure 5D*). However, there was no increase in CD247 expression upon stimulation with *S. aureus* in the *Terc*ko/ko T cells (*Figure 5D*). In contrast, observation of T cells of young WT mice revealed a tendency toward

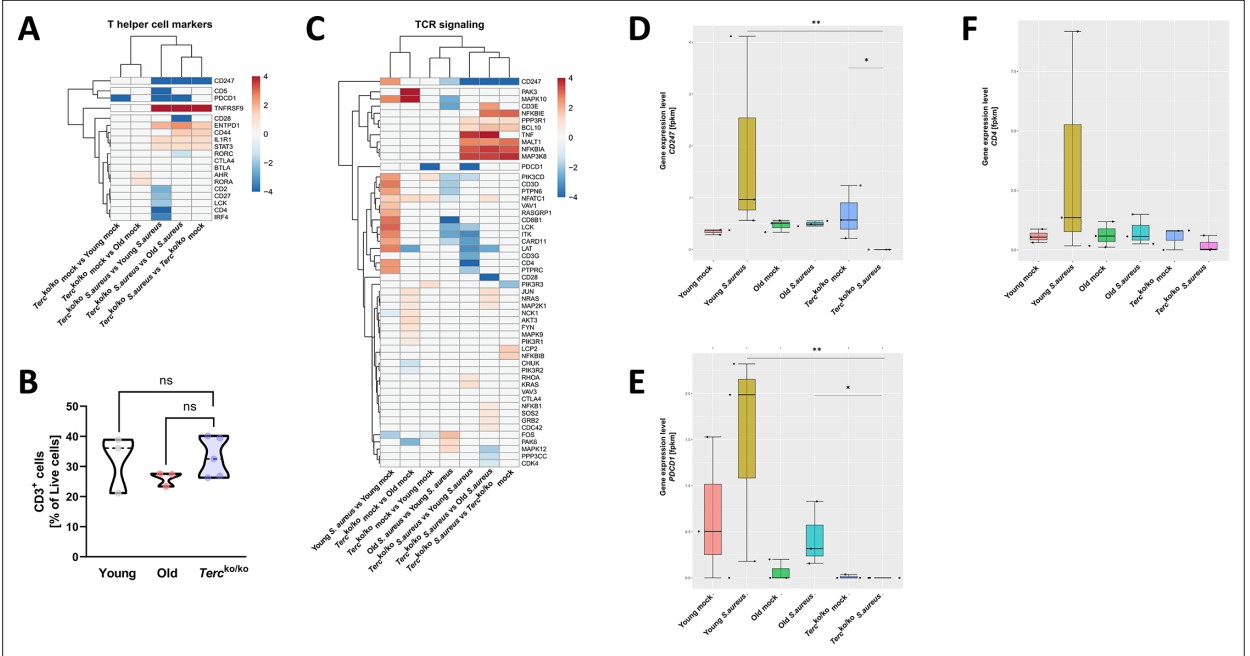

**Figure 4.** T cell receptor (TCR) in infected *Terc*ko/ko mice is partially downregulated. Heatmap of differentially expressed T helper cell markers (**A**) and differentially expressed genes (DEGs) belonging to the TCR signaling pathway (**C**). Red indicates upregulation, and blue indicates downregulation of the gene. Differential gene expression analysis was conducted using the DESeq2 software as well as negative binomial distribution. For false discovery rate (FDR) calculation the Benjamini-Hochberg procedure was used. Heatmaps were constructed using RStudio. Only significant DEGs (p-value<0.05) were displayed in the heatmap. Non-significant genes were set to zero and are shown as white. DEGs are displayed as log2fold-change. For each mouse cohort three biological replicates were sequenced and used for analysis. The respective groups used for the comparison are indicated below each column. All heatmaps display mRNA sequencing data of non-infected and infected lungs of the three different mice cohorts at 24 hr post infection (hpi). (**B**) Percentage of CD3⁺-living cells in non-infected young WT (n=3), old WT (n=3), and *Terc*ko/ko mice (n=5) spleen. Cells were analyzed with flow cytometry. Gene expression levels in fragments per kilobase of transcript per million fragments mapped (fpkm) of *CD247* (**D**), *PDCD1* (**E**), and *CD4* (**F**) in the lungs of non-infected and infected young WT (n=3), old WT (n=3), and *Terc*ko/ko mice (n=3) at 24 hpi. ns≥0.05, *p<0.05, **p<0.01. p-Values calculated by Kruskal-Wallis test (**B, D–F**). Data is displayed as violin or box plot showing the median as well as lower and upper percentile of each dataset. Each replicate is displayed as a data point.

The online version of this article includes the following source data and figure supplement(s) for figure 4:

**Source data 1.** Numeric data for *Figure 4A*.

**Source data 2.** Numeric data for *Figure 4B*.

**Source data 3.** Numeric data for *Figure 4C*.

**Source data 4.** Numeric data for *Figure 4D–F*.

**Source data 5.** List of all fragments per kilobase of transcript per million fragments mapped (fpkms) for each individual mouse sequenced.

**Figure supplement 1.** Frequency of different immune cell populations in the spleen of young and old WT as well as *Terc*ko/ko mice.

**Figure supplement 1—source data 1.** Numeric data for *Figure 4—figure supplement 1B–F*.

an increase in CD247 expression in response to *S. aureus* stimulation (*Figure 5D*). This indicated improper activation of TCR signaling after *S. aureus* stimulation in T cells of *Terc*ko/ko mice.

We could also detect differences in cytokine production when comparing the AMs and T cells from young WT and *Terc*ko/ko mice. At 24 hpi, IL-1α production was significantly higher in T cells from young WT mice stimulated with *S. aureus* when compared to the T cells from *Terc*ko/ko mice (*Figure 5D*, left). Notably, in the co-culture, IL-1α production was significantly higher in *Terc*ko/ko cells 2 hpi (*Figure 5D*, right). On the contrary, there was a significant induction of IL-1α in the young WT co-culture but no increase in the *Terc*ko/ko co-culture with *S. aureus* stimulation after 24 hr (*Figure 5D*, right).

Our data indicate that T cells from *Terc*ko/ko mice do not effectively react to the infectious stimulus and thus seem dysfunctional due to TCR downregulation.

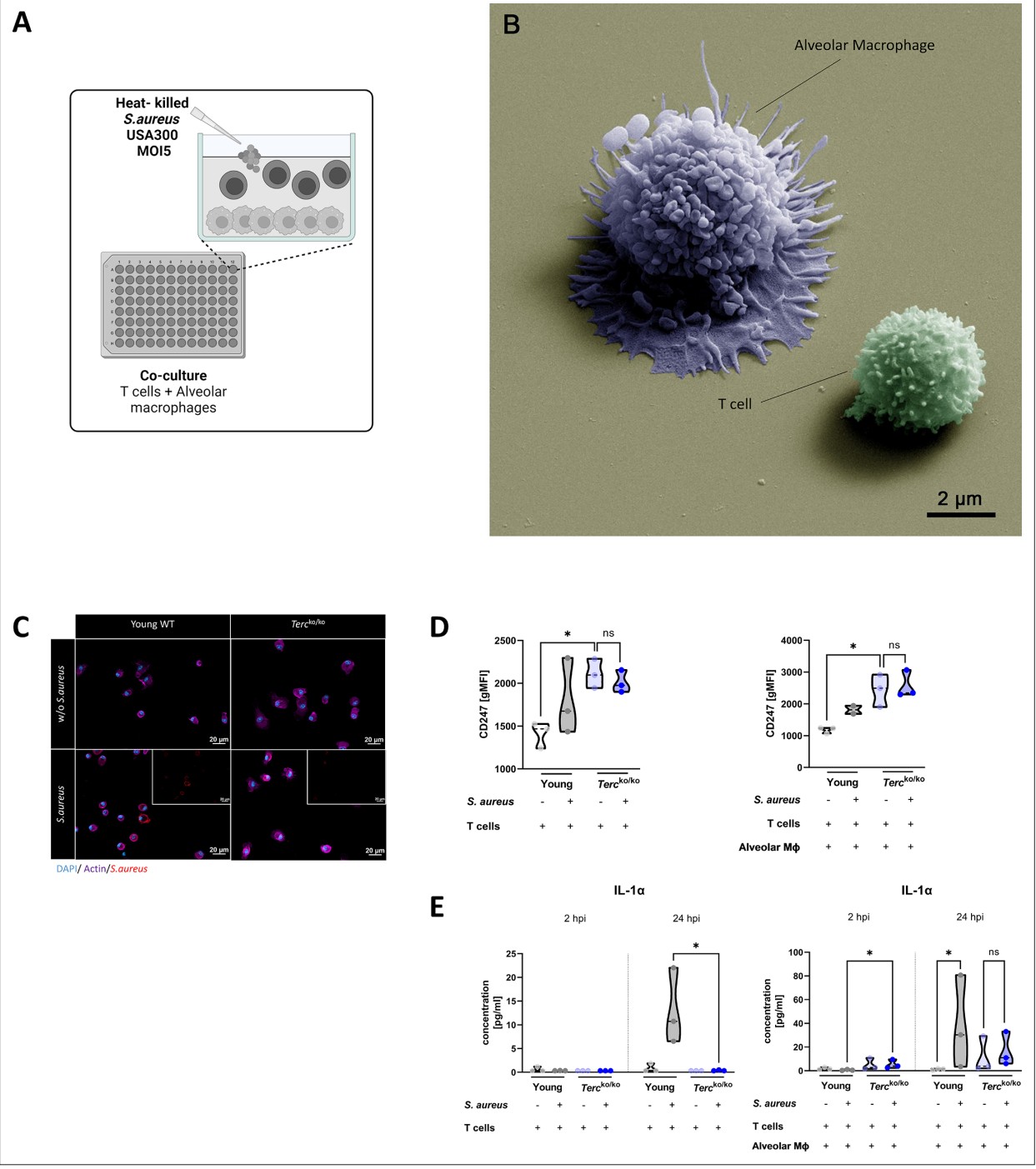

**Figure 5.** T cells of *Terc*ko/ko mice do not adequately react to bacterial challenge. (**A**) Schematic representation of experiment. Murine T cells and alveolar macrophages (AMs) were isolated, co-cultured, and stimulated with heat-killed *S. aureus* with a multiplicity of infection 5 (MOI5) for 24 hr. (**B**) Colorized scanning electron microscopy (SEM) picture of the experimental setup. An AM close to a T cell is shown. (**C**) Immunofluorescence staining of infected and non-infected AMs from young wild-type (WT) and *Terc*ko/ko mice. AMs were stained with antibodies for *S. aureus* (red), actin (purple), and DAPI (blue). (**D**) CD247 expression measured with flow cytometry of non-infected and infected T cells (left) and co-cultured T cells and AMs (right) of young WT (each n=3) and *Terc*ko/ko mice (each n=3) after 24 hr post infection (hpi). CD247 expression is displayed as geometric mean fluorescence intensity (gMFI). (**E**) Interleukin-1α (IL-1α) from supernatants of T cells (left) and co-cultured T cells and AMs (right) of young WT (each n=3) and *Terc*ko/ko mice (each n=3) at 2 and 24 hpi. IL-1α levels were measured using a flow cytometry-based LEGENDPlex mouse T helper cytokine panel. ns≥0.05, *p<0.05. p-Values calculated by Kruskal-Wallis test (**D, E**). Data is displayed as violin plot showing the median as well as lower and upper percentile of each dataset. Each replicate is displayed as a data point. Created with BioRender.com.

*Figure 5 continued on next page*

*Figure 5 continued*

The online version of this article includes the following source data and figure supplement(s) for figure 5:

**Source data 1.** Numeric data for *Figure 5D*.

**Source data 2.** Numeric data for *Figure 5E*.

**Figure supplement 1.** Altered immune response to infection of *Terc*ko/ko mice relative to control cohorts.

**Figure supplement 1—source data 1.** Numeric data for *Figure 5—figure supplement 1A*.

**Figure supplement 1—source data 2.** Numeric data for *Figure 5—figure supplement 1A*.

**Figure supplement 1—source data 3.** Numeric data for *Figure 5—figure supplement 1A*.

**Figure supplement 1—source data 4.** Numeric data for *Figure 5—figure supplement 1C*.

**Figure supplement 1—source data 5.** Numeric data for *Figure 5—figure supplement 1D*.

## Discussion

Shortening of telomeres is a key factor in the aging process and has a significant impact on other hallmarks of aging, such as mitochondrial dysfunction and cellular senescence (*Birch et al., 2018*; *López-Otín et al., 2023*). In this context, the discussion surrounding the importance of *Terc* is primarily confined to its role within telomerase. Consequently, other potential functions of Terc in various cellular processes have been largely overlooked. Interestingly, various studies have linked Terc to inflammation, as it has been shown to activate the NF-κB and PI3K-AKT pathways (*Liu et al., 2019*; *Wu et al., 2022*). Another study could show that the deletion of *Terc* in mice resulted in NLRP3 inflammasome activation after infection with *S. aureus*, thus contributing to more severe pneumonia (*Kang et al., 2018*). These studies suggest a close association between Terc, inflammation, and the response to bacterial lung infections.

In our study, we utilized a *Terc*ko/ko *S. aureus* pneumonia model to investigate this connection. In accordance with German animal welfare regulations and the 3R principle, we only had a reduced number of *Terc*ko/ko mice available for the experiment, as some infections were fatal. This reduced number of mice is a limitation of our study. However, we could demonstrate that *Terc*ko/ko mice presented with a more severe pneumonia, higher bacterial load, and increased mortality compared to old and young WT mice (*Figure 1*). Additionally, we could show that infected *Terc*ko/ko mice are rather heterogeneous and display several degrees of severity, which we investigated separately. Of five infected mice, two died, two displayed signs of a systemic infection and only one had a mild infection. The mice with systemic infection were grouped based on the presence of bacteria in organs other than the lung, as well as the sequencing data, where they clustered separately. The detection of bacteria in extrapulmonary organs is of particular interest, as bacteremia is a symptom of severe pneumonia and is associated with high mortality (*De la Calle et al., 2016*). The grouping of *Terc*koko mice, based on their disease severity, offers the possibility to investigate distinct features of bacterial infection. *Terc*ko/ko mice were previously shown to have higher susceptibility to *S. aureus* pneumonia, although they exhibited a reduced bacterial load in their lungs (*Kang et al., 2018*). These differences may be attributed to the missing separation of infected *Terc*ko/ko mice into different degrees of severity. Additionally, this heterogeneity in disease manifestation points toward the limitations of mouse models to study *S. aureus* pathologies. There are a variety of differences between *S. aureus* infections in humans and mice. For instance, several relevant virulence factors (e.g. Pantone-Valentine leukocidine) function less efficient in mice than in humans (*Zheng et al., 2023*). Therefore, higher infection doses are needed to successfully establish an infection that is comparable to humans (*Zheng et al., 2023*).

Inflammation and the activation of the NLRP3 inflammasome are necessary components to fight bacterial infections of the lung. However, excessive inflammation and over-activation of the NLRP3 inflammasome lead to lung injury and can impede pathogen clearance, thus contributing to *S. aureus*-induced pathologies (*McVey et al., 2021*; *Wang et al., 2022*). Upon examining the lungs of infected *Terc*ko/ko mice with systemic infection, we identified a marked increase in inflammation parameters (*Figure 2*). Furthermore, we discovered that genes associated with the NLRP3 inflammasome pathway were elevated in the infected *Terc*ko/ko mice but not in the corresponding non-infected control cohort. As published previously, this implies that *Terc* knockout on its own does not result in heightened inflammation and inflammasome activation but rather requires an external trigger (*Kang et al., 2018*). Inflammasome activation in *Terc*ko/ko mice was connected to mitochondria dysfunction

and an elevated reactive oxygen species production. Additionally, regulation of the inflammasome seems to be dysfunctional as TNAIFP3, an important negative regulator of the inflammasome was downregulated in the $Terc^{ko/ko}$ mice in this study (*Kang et al., 2018*). However, we did not observe downregulation of TNFAIP3, but rather an upregulation in infected $Terc^{ko/ko}$ mice (*Figure 2C*). These differences may however be attributed to the different study design. Kang et al. focused on telomere dysfunction studying macrophages of old $Terc^{ko/ko}$ mice, while our study investigated the entire lung of young $Terc^{ko/ko}$ mice. Of particular note were the heightened levels of IL-1β, a factor known for its ability to amplify the inflammatory response and facilitate the infiltration of pro-inflammatory immune cells, thereby fostering pulmonary injury (*Chen et al., 2018*; *Hu et al., 2020*). Additionally, pro-inflammatory macrophages infiltrating the lung are critical to inflammation-mediated lung injury (*Chen et al., 2018*; *Guan et al., 2023*). In our study, we could identify the upregulation of several pro-inflammatory macrophage markers in infected $Terc^{ko/ko}$ mice, as well as several chemokine markers further exacerbating the inflammation in the lung via attraction of immune cells (*Figure 3*). Notably, we did not observe upregulation of inflammation parameters or activation of the inflammasome in the infected old WT mice. Although *CD14* showed a slight increase in non-infected old WT mice compared to non-infected young WT mice, there was no upregulation of macrophage markers in the infected cohorts of these groups. Innate immune cells of aged individuals especially macrophages are associated with chronic low-grade inflammation contributing to injury during infection (*Duong et al., 2021*). However, studies could also show that macrophages of aged mice had a reduced response to stimulation of pathogen recognition receptors such as Toll-like receptor 2. This results in an initially diminished and delayed response of macrophages to infectious challenge, which could be a possible explanation for the lack of inflammation at 24 hpi (*Boe et al., 2017*; *Hinojosa et al., 2009*).

An overactive inflammatory response not only causes tissue damage but also leads to T cell dysfunction (*Guan et al., 2023*). A possible mechanism is the downregulation of CD247, which is the CD3ζ chain of the TCR. CD247/CD3ζ is necessary for the assembly of the receptor at the cell surface. Furthermore, due to its multiple immunoreceptor tyrosine-based activation motifs, it is a central factor in the signal transduction (*Bronstein-Sitton et al., 2003*; *Jin et al., 2024*). The downregulation or absence of CD247/CD3ζ is linked to reduced responsiveness in T cells and is commonly present in autoimmune diseases like rheumatoid arthritis and lupus (*Dexiu et al., 2022*). This mechanism also occurs in various chronic infectious diseases, including HIV and hepatitis C (*Dexiu et al., 2022*). However, it can also be a general regulatory process in T cells during infection, helping to maintain immune homeostasis and prevent excessive inflammation (*Baniyash, 2004*; *van der Donk et al., 2021*). Thus, the connection between CD247/CD3ζ downregulation during chronic inflammation and chronic infectious diseases has been well documented (*Dexiu et al., 2022*). Interestingly, we could show that expression of *CD247* is entirely absent in infected $Terc^{ko/ko}$ mice, while in the non-infected $Terc^{ko/ko}$ mice expression of *CD247* could be detected after 24 hr of infection. Hence, our data indicates that infection combined with deletion of *Terc* is associated with reduction of *CD247* expression. Our study provides the first evidence of the total absence of *CD247* expression during acute bacterial infection and inflammation (*Figure 4*).

Previous work has already reported that constant stimulus with bacterial antigens reduces the expression of CD247/CD3ζ and impairs T cell function (*Bronstein-Sitton et al., 2003*). However, the complete absence of CD247/CD3ζ after a single antigenic stimulus, without chronic inflammation as well as the connection to knockout of *Terc*, are novel findings. These findings require further studies and investigations to gain a deeper understanding of the underlying mechanisms.

Downregulation of CD247/CD3ζ is facilitated by myeloid suppressor cells (MSCs), which accumulate at the site of inflammation (*Baniyash, 2004*; *Dexiu et al., 2022*; *Ezernitchi et al., 2006*). MSCs are cells which exert an immunosuppressive function on T cells in response to inflammation or infection (*Medina and Hartl, 2018*). Thus MSCs contribute toward the regulation of the immune response to pathogens to prevent an overshooting reaction of the immune system (*Medina and Hartl, 2018*). However, their immunosuppressive function can also exacerbate infections. For instance, elevated levels of MSC induced T cell dysfunctionality and caused an exacerbated and sustained chronic infection in an *S. aureus* mouse model (*Tebartz et al., 2015*). Immunosuppression of T cells is conveyed by multiple mechanisms, one of which is the downregulation of CD247/CD3ζ (*Ezernitchi et al., 2006*). This process is facilitated among others by the ability of MSCs to metabolize the amino acid arginine, as arginine seems to be essential for expression of CD247 in T cells (*Ezernitchi et al., 2006*;

*Rodriguez et al., 2004*). Interestingly, we could identify several MSC markers, such as lymphocyte antigen 6 family member G (*LY6G*), to be upregulated in the infected *Terc*ko/ko mice. This suggests a potential role of MSCs in the downregulation of CD247/CD3 ζ (*Figure 5—figure supplement 1D*). Notably, we could identify reduced expression of *CD247* in infected old WT mice compared to young WT mice. Moreover, our sequencing data did not show an increase in *CD247* expression in infected old WT mice compared to the non-infected control group, which further suggests that the T cells of old WT mice may also exhibit some level of dysfunction (*Figure 4*). One potential underlying mechanism is the reduced ability of aged macrophages to activate T cells due to diminished expression of the receptor required for presentation of the antigen (*Herrero et al., 2001*). Furthermore, T cells of aged individuals become increasingly deficient due to immunosenescence and exhaustion (*Mittelbrunn and Kroemer, 2021*). Supporting this, we could observe downregulation of *CD27* compared to infected young WT mice, while *CTLA4* was upregulated in infected old WT compared to uninfected mice (*Figure 2—figure supplement 1E and F*). Downregulation of CD28 and CD27 is part of T cell aging and an essential marker of T cell senescence, while upregulation of CTLA4 is part of a signature pointing to T cell exhaustion (*Zhao et al., 2020*). Interestingly, senescent T cells have shorter telomeres and exhibit reduced telomerase activity, which indicates a possible connection between Terc and T cell function (*Mittelbrunn and Kroemer, 2021*). Interestingly, *CD28* expression was downregulated in infected *Terc*ko/ko mice compared to old WT infected mice (*Figure 4A and C*). However, we did not observe any other relevant changes of T cell senescence or exhaustion markers in infected *Terc*ko/ko mice (*Figure 4A and C*). Thus, T cell senescence could be a contributing factor to the T cell dysfunctionality, but likely does not explain the complete phenotype. The differences in the expression of T cell senescence and exhaustion markers, in addition to the lack of increased inflammation in old WT mice, indicate that mechanisms causing downregulation of parts of the TCR and T helper cell markers in old WT and *Terc*ko/ko mice differ from each other. As T cell senescence is however a known consequence of *Terc* deletion, the lack of in depth investigation of this process in the present study is a limitation (*Matthe et al., 2022*). It is important to note that telomere shortening has a significant impact on the immune system (*Chakravarti et al., 2021*). Although young *Terc*ko/ko mice were used in this study, telomere shortening is still likely to be a contributing factor. Therefore, further experiments investigating the role of T cell senescence in this model should be conducted.

Nevertheless, old WT mice exhibit certain immune cell dysfunction similarities. However, given the intricate nature of aging, other mechanisms contribute to the distinct response observed in *Terc*ko/ko and old WT mice following infection. This highlights the necessity of employing a knockout model to specifically examine the influence of Terc on infection, isolating it from the multifaceted processes involved in aging.

We observed an increase in pro-inflammatory macrophages in the lungs of *Terc*ko/ko mice during infection. To explore the role of macrophages in regulating T cell function during infection, we investigated whether AMs were sufficient to induce CD247/CD3 ζ downregulation in T cells. While CD247/CD3 ζ expression increased with bacterial stimulation in T cells from young WT mice, we failed to induce such an increase in CD247/CD3 ζ expression in T cells from *Terc*ko/ko mice (*Figure 5*). The expression pattern of CD247/CD3 ζ was similar to that observed in the sequencing data, as initial levels of CD247/CD3 ζ expression in non-infected *Terc*ko/ko mice were higher than those of young WT mice. However, there was no significant reduction of CD247/CD3 ζ after stimulation with bacteria in the T cells of *Terc*ko/ko mice. One potential explanation for this finding is the absence of MSCs and other infiltrating cells, which can contribute to the inflammatory environment and lead to T cell dysfunction. Furthermore, after 24 but not 2 hr of infection, we observed a reduced release of the pro-inflammatory cytokine IL-1α from *Terc*ko/ko T cells, suggesting a time-dependent reduction of T cell function. The results of the experiment conducted on T cells of *Terc*ko/ko mice revealed an insufficient response to the bacterial stimulation, as there was no significant increase in CD247/CD3 ζ expression or IL-1α release after 24 hpi. As total CD4⁺ T cells were analyzed in this study, it would be useful to investigate specific T cell populations such as memory and effector T cells to elucidate the potential mechanism leading to T cell dysfunctionality in further detail. Additionally, analysis of differences in immune cell recruitment to the lungs between young WT and *Terc*ko/ko mice would be relevant. Thus, additional experiments are needed to validate and further investigate the underlying mechanisms leading to T cell dysfunctionality in *Terc*ko/ko mice.

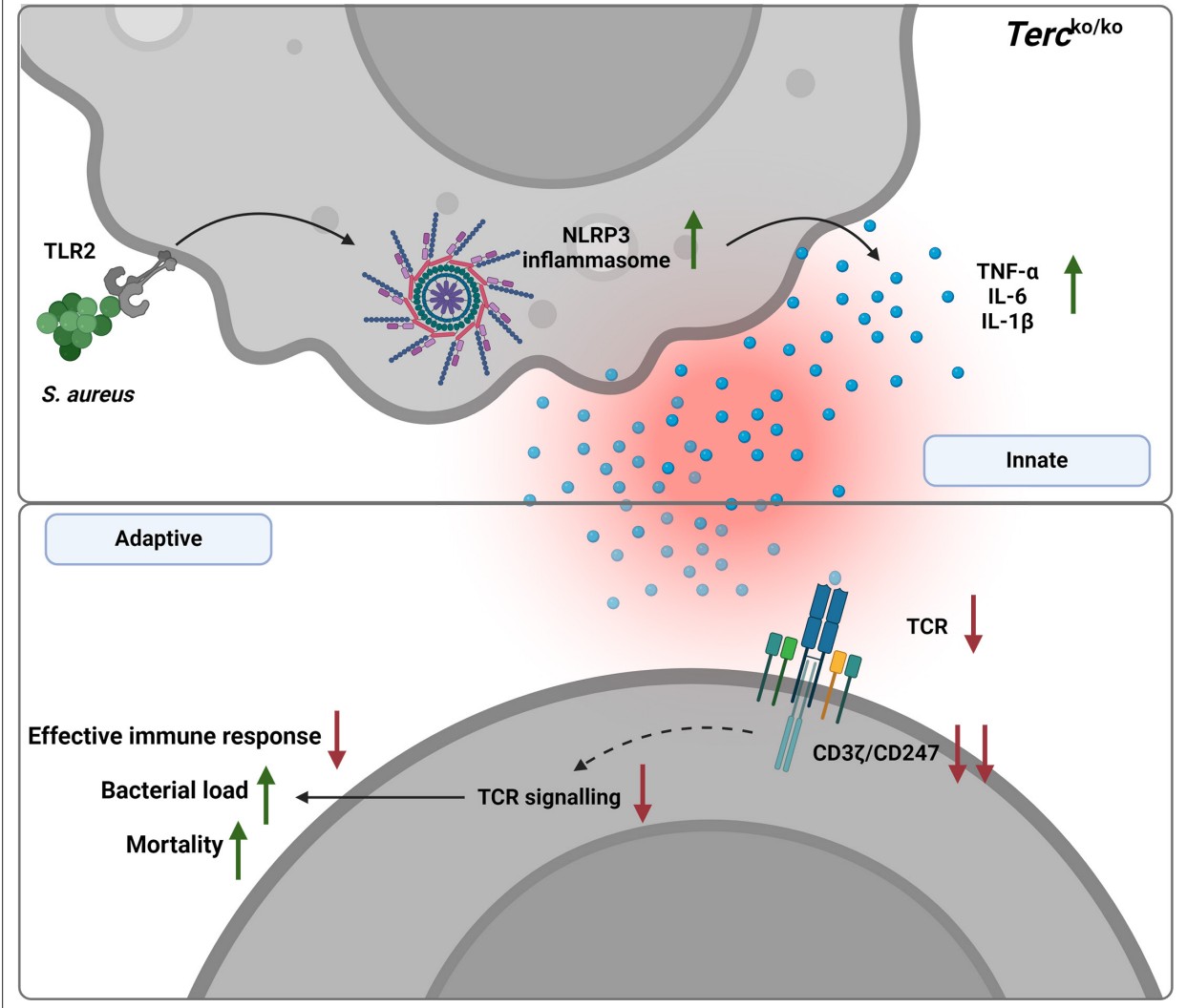

**Figure 6.** Graphical summary. Loss of *Terc* leads to an exaggerated inflammatory response and inflammasome activation upon *S. aureus* infection. This overactive inflammation disrupts T cell function, thereby impairing the immune response. Consequently, *Terc*^ko/ko^ mice show a significantly higher bacterial load and increased mortality following infection. Created with BioRender.com.

The online version of this article includes the following figure supplement(s) for figure 6:

**Figure supplement 1.** Overview of the methods used in this publication.

Our data points toward the pivotal role of Terc in the regulation of inflammation as well as maintaining immune homeostasis during infection. We could demonstrate that *S. aureus* pneumonia in *Terc*^ko/ko^ mice resulted in an overshooting inflammation and activation of the NLRP3 inflammasome. The elevated inflammation eventually leads to T cell dysfunction, which impairs the host's ability to mount an effective immune response against the pathogen, resulting in increased bacterial load and mortality. As T cell dysfunctionality is caused by the excessive inflammation, other pathogens inducing high inflammatory responses in the host could also lead to a similar phenotype in *Terc*^ko/ko^ mice (**Figure 6**).

This study linked T cell dysfunction to *Terc* deletion, which is supported by previous studies reporting a connection between reduced expression of *Terc* and apoptosis in human CD4⁺ T cells (**Gazzaniga and Blackburn, 2014**). These findings provide insights into possible mechanisms contributing to a more severe *S. aureus* pneumonia in the aging population.

# Materials and methods

## Bacteria

All in vivo and in vitro experiments were carried out with the *S. aureus* strain USA300. *S. aureus* was cultivated in brain heart infusion medium (Thermo Fisher Scientific, Waltham, MA, USA) at 37°C while shaking at 150 rpm. The bacteria were harvested during the mid-logarithmic phase and adjusted to the appropriate optical density at 600 nm for the respective experiments in phosphate-buffered saline (PBS). The bacteria were then either used freshly for in vivo infection of mice or stored at –80°C until further usage in in vitro experiments.

For heat-killing, the adjusted bacteria stock was incubated at 95°C for 5 min while shaking at 500 rpm. A fraction of the HK bacteria was plated on Müller-Hinton (MH) plates. Plates were incubated for 5 days at 37°C to confirm heat-killing of the bacteria.

## Determination of bacterial load

Organs were homogenized in the appropriate amount of PBS (adjusted to their weight in mg) using a SpeedMill Plus (Analytik Jena, Jena, Germany). Homogenized organs and BAL fluid were diluted in a decadic dilution series and plated onto MH plates. Plates were incubated overnight at 37°C. Colonies were counted, and CFU per ml was calculated.

## Mouse models and infection

Mouse in vivo experiments were approved by the Office for Consumer Protection of Thuringia (TV-Number: UKJ-19-028 and UKJ-22-023).

G3 female *Terc*[ko/ko] mice with a C57Bl/6 background were bred from homozygous parents in the animal facility of the Jena University Hospital. To confirm knockout of *Terc*, DNA was extracted from the tail of newborn mice using the DNeasy Blood & Tissue Kit (QIAGEN, Venlo, Netherlands) according to the manufacturer's protocol. The DNA was then used for genotyping via PCR using the following primer sequences: mTR-R: 5'-TTC TGA CCA CCA CCC ACT TCA AT-3'; mTR-WT-F: 5'-CTA AGC CGG CAC TCC TTA CAA G-3'; 5PPgK(-F): 5'-GGG GCT GCT AAA GCG CAT-3'. The results were analyzed with an Agilent 2200 TapeStation system using a Agilent D1000 ScreenTape (both Agilent Technologies, Santa Clara, CA, USA). The WT *Terc* product had a size of 200 bp while the knockout product had a size of 180 bp. Only mice that showed a single band at 180 bp were used for the further breeding process.

*Terc*[ko/ko] mice aged 8 weeks were used for infection studies (n=8; non-infected=3; infected = 5). Female young WT (age 8 weeks) and old WT (age 24 months) C57Bl/6 mice (both n=10; non-infected=5; infected = 5) were purchased from Janvier Labs (Le Genest-Saint-Isle, France). All infected mouse cohorts were compared to their respective non-infected controls, as well as to the infected groups from other cohorts. Additionally, comparisons were made between the non-infected cohorts across all groups.

All mice were anesthetized with 2% isoflurane before intranasal infection with *S. aureus* USA300 ($1\times10^8$ CFU/20 µl) per mouse. After 24 hr, the mice were weighed and scored as previously described (*Hornung et al., 2023*). Infected Terc[ko/ko] mice were grouped into different degrees of severity based on their clinical score, fatal outcome of the disease (fatal), and the presence of bacteria in organs other than the lung (systemic infection) for the indicated analysis. Mice with fatal infections were excluded from subsequent analyses, with only their final scores being reported. The mice were sacrificed via injection of an overdose of xylazine/ketamine and bleeding of axillary artery after 24 hpi. BAL was collected by instillation and subsequent retrieval of PBS into the lungs. Serum and organs were collected. Bacterial load in the BAL, kidney, and liver was determined by plating of serially diluted sample as described above. For this organs were previously homogenized in the appropriate volume of PBS. Gene expression was analyzed in the right superior lung lobe. Lobes were therefore homogenized in the appropriate amount of TRIzol LS reagent (Thermo Fisher Scientific, Waltham, MA, USA) prior to RNA extraction. The left lung lobe was embedded into Tissue Tek O.C.T. (Science Services, Munich, Germany) and stored at –80°C until further processing for histological analysis. Cytokine measurements were performed using the right inferior lung lobe. Lobes were previously homogenized in the appropriate volume of PBS. Remaining organs were stored at –80°C until further usage.

Mouse studies were conducted without the use of randomization or blinding.

## Telomere length measurement

The length of telomeres in the lung of young WT, old WT, and $Terc^{ko/ko}$ mice was measured using the Absolute Mouse Telomere Length Quantification qPCR Assay Kit (ScienCell Research Laboratories, Carlsbad, CA, USA) according to the manufacturer's protocol. Genomic DNA from the lungs was extracted by DNeasy Blood & Tissue Kit (QIAGEN, Venlo, Netherlands) according to the manufacturer's protocol. For the PCR 2 ng of each sample was used. Average telomere length per chromosome end was calculated as described by the manufacturer.

## Histology and immunofluorescence

Preparation and staining of tissue cryosections was performed as described previously (*Hornung et al., 2023*).

For staining of *S. aureus,* rabbit anti- *S. aureus* polyclonal antibody (Cat. #PA1-7246, Thermo Fisher Scientific, Waltham, MA, USA) was diluted 1:400 in antibody diluent with background reducing components (Agilent, Santa Clara, CA, USA). Alexa Fluor 488 AffiniPure Donkey Anti-Rabbit IgG (Cat. #711-545-152, Jackson ImmunoResearch, West Grove, PA, USA) and BODIPY 558/568 phalloidin, which stains actin fibers (Cat. #B3475, Thermo Fisher Scientific, Waltham, MA, USA) were diluted in antibody diluent 1:500 and 1:400 respectively and applied to the slides as secondary antibodies.

## Immunocytochemistry

Murine primary macrophages were fixed with 4% paraformaldehyde (PFA) for 15 min at 37°C, subsequently permeabilized with 0.1% Triton X-100, and blocked with 3% bovine serum albumin (BSA) in PBS for each 30 min at room temperature (RT). Cells were incubated overnight with either rabbit anti- *S. aureus* polyclonal antibody or rabbit anti-CD68 polyclonal antibody (Cat. # BOSSBS-0649R, VWR, Radnor, PA, USA) diluted 1:400 or 1:200 in 3% BSA in PBS, respectively. The following day, the slides were washed three times with PBS. Cy3-conjugated AffiniPure Donkey Anti-Rabbit IgG (Cat. #711-165-152, Jackson ImmunoResearch, West Grove, PA, USA) and Alexa Fluor Plus 647 Phalloidin (Cat. # A30107, Thermo Fisher Scientific, Waltham, MA, USA) were applied to the cells diluted 1:500 and 1:400 in 3% BSA in PBS respectively and incubated for 1 hr at RT. The cells were then mounted with DAPI Fluoromount-G.

The immunofluorescence pictures were taken with an AxioObserver Z.1 microscope (Carl Zeiss AG, Oberkochen, Germany) and analyzed with the Zen software (Zen Pro v3.3).

## RNA extraction and mRNA sequencing

RNA was extracted from murine lung tissue using the TRIzol/chloroform method. First, about 15 mg of lung tissue was homogenized in TRIzol LS reagent (Thermo Fisher Scientific, Waltham, MA, USA) using the SpeedMill Plus. The tissue-free supernatant of the homogenate was transferred into a new tube after centrifugation. 50 µl of chloroform was added to 250 µl TRIzol Reagent, incubated for 3 min at RT, and centrifuged at 12,000×$g$ for 15 min at 4°C. The aqueous phase was mixed with isopropanol in a 1:2 ratio and incubated at RT for 10 min before centrifugation at 12,000×$g$ for 10 min at 4°C. Afterward, the RNA pellet was washed twice with 75% ethanol. Finally, ethanol was removed, and the pellet was air-dried for 5–10 min. The dry pellet was then dissolved in sterile distilled water, and RNA concentration was determined by an ND-1000 NanoDrop spectrophotometer (PEQLAB Biotechnologie GmbH, Erlangen, Germany). Before sequencing, RNA integrity was measured using a 5400 Fragment Analyzer (Agilent Technologies, Santa Clara, CA, USA). RNA concentration and RNA integrity of the samples can be found in *Supplementary file 1*.

Library construction and mRNA sequencing were performed by Novogene Co., LTD. (Beijing, China) using the Illumina platform NovaSeq 6000 S4 flowcell v1.0, based on the sequencing by synthesis mechanism and PE150 strategy. For bioinformatics analysis, raw reads in FASTQ format were first processed using fastp. Reads containing adapters and poly N sequences were removed as well as low quality reads to generate clean data. Clean reads were then mapped to the reference genome (*Mus musculus* [GRCm39/mm39]) with the HISAT2 software. Gene expression levels were quantified using the fpkm (fragments per kilobase of transcript per million fragments mapped) method. DESeq2 software as well as negative binomial distribution was used for analysis of differential gene expression. For FDR (false discovery rate) calculation the Benjamini-Hochberg procedure was used. A summary of the quality of the sequencing data of each sample can be found in *Supplementary file 2*.

Heatmaps of the differentially expressed genes (DEGs) were constructed using R (v4.2.2; R Foundation for Statistical Computing, https://www.r-project.org/). Only significant DEGs (p-value<0.05) were displayed in the heatmaps. Non-significant genes were set to zero and are shown in white. All DEGs used for construction of the respective heatmaps are summarized in *Supplementary file 3a-i* .

## Immunophenotyping of the spleen

Mice were sacrificed as described above, and spleens were freshly stored in cold 2% fetal calf serum (FCS) in PBS. For isolation of immune cells, spleens were mashed through a 70 µm cell strainer (Miltenyi Biotec, Bergisch Gladbach, Germany), and red blood cell lysis was performed by adding 5 ml of ammonium-chloride-potassium lysis buffer (Thermo Fisher Scientific, Waltham, MA, USA) for 5 min at 4°C to the pelleted cells. Cells were then counted, adjusted to $2 \times 10^6$ per ml PBS, and stained with BD Fixable Viability Stain 780 (BD Biosciences, Heidelberg, Germany) diluted 1:1000 in PBS for 15 min at RT. Non-specific binding of antibodies was blocked by incubating the samples with 3% rat serum in BD staining buffer (BD Biosciences, Heidelberg, Germany) for 5 min at RT. Anti-CD11c (Cat. #557401), anti-CD19 (Cat. #566412), anti-CD335 (Cat. #564069), anti-CD34 (Cat. #742971), anti-CD3ε (Cat. #553066), anti-Ly6G (Cat. #560601), and anti-F4/80 (Cat. #565411) were added to the samples in a dilution of 1:100 and incubated for 20 min at 4°C. All antibodies were purchased from BD Biosciences. Stained samples were measured with a BD Symphony A1 (BD Biosciences, Heidelberg, Germany), and analysis was performed with FlowJo v10.8.1. Detailed information on the gating strategy is presented in *Figure 4—figure supplement 1A*.

## Isolation and infection of murine T cells and AMs

*Terc*ko/ko and young WT mice were sacrificed. Killing for scientific purposes was conducted in accordance with the German animal welfare regulations. AMs were isolated as published elsewhere (*Rios et al., 2017*). In short, the murine lungs were washed out several times with PBS containing 0.5 mM ethylenediaminetetraacetic acid. The lavage was centrifuged at 350×*g* for 10 min at 4°C. The cell pellet was dissolved in Roswell Park Memorial Institute (RPMI) medium containing 20 mM HEPES and 5% FCS and counted. Subsequently, cells were seeded and incubated at 37°C for 2 hr.

Lymphocytes were isolated from the spleen as described above. To subsequently isolate CD4+ T cells, cells were separated via magnetic cell separation. These cells were mixed with CD4+ beads and separated via MACS LS Columns (both Miltenyi Biotec, Bergisch Gladbach, Germany) according to the manufacturer's protocol. The purity of the resulting CD4+ T cell population was measured via flow cytometry using a BD Symphony A1 (BD Biosciences, Heidelberg, Germany). T cells were added to AMs for co-culture in a 1:1 ratio or cultured alone.

For infection with HK *S. aureus,* an MOI of 5 was calculated, and the appropriate bacterial volume was added to each well. Cells were then incubated for 24 hr and harvested for downstream analysis.

## Cytokine measurement

For cytokine measurements, the right inferior lung lobe was homogenized using a SpeedMill Plus in the appropriate volume of PBS to adjust the organs to their weight in mg. Then, the cytokines were measured with the LEGENDPlex mouse inflammation panel (BioLegend, San Diego, CA, USA) according to the manufacturer's instructions. Samples were run in duplicates on the same plate. For the young WT and *Terc*ko/ko cohort three biological replicates and for old WT cohorts five biological replicates were measured. Supernatants of co-cultured and single-cultured T cells were measured with the LEGENDPlex mouse T helper cytokine panel (BioLegend, San Diego, CA, USA). For each condition three biological replicates were measured. All samples were run on the same plate. Flow cytometry analysis was performed with a BD Symphony A1, and data analysis was carried out with the Qognit software v2023-02-15 (BD Biosciences, Heidelberg, Germany).

## Flow cytometry

Co-cultured and single-cultured T cells were transferred to FACS tubes and centrifuged at 500×*g* for 5 min at 4°C. The supernatant was frozen for cytokine measurements, and the cell pellet was stained with PBS containing 1:1000 diluted BD Fixable Viability Stain 780 (Cat. #565388, BD Biosciences, Heidelberg, Germany) for 15 min at RT. Staining of surface markers was performed as described above. Samples were stained with anti-CD3ε (1:100, Cat. #553066), anti-CD25 (1:800, Cat. #553866),

anti-CD69 (1:400, Cat. #551113), and anti-CD44 (1:100, Cat. #560569) for 30 min at 4°C. All antibodies were purchased from BD Pharmingen. Subsequently, samples were permeabilized and fixed. For this, the samples were centrifuged for 5 min at 500×$g$ and 4°C, resuspended in BD Cytofix/Cytoperm buffer (BD Biosciences, Heidelberg, Germany), and incubated for 20 min at 4°C. Samples were washed once with BD Perm/Wash Buffer (BD Biosciences, Heidelberg, Germany). For intracellular staining, samples were stained with anti-CD247 (1:100, Cat. #ab91493, Abcam, Cambridge, UK) antibody in BD Perm/Wash Buffer for 30 min at 4°C. Lastly, the samples were washed and resuspended in BD Perm/Wash Buffer. Stained samples were measured with a BD Symphony A1 and analyzed with FlowJo v10.8.1. Detailed information on the gating strategy is presented in *Figure 5—figure supplement 1B*.

## Scanning electron microscopy

T cells and AMs were isolated as described above. Cells were grown on poly-L-lysine (Merck Millipore, Burlington, MA, USA) coated glass coverslips in a 1:1 ratio. T cells were allowed to attach to the coverslips for 30 min before adding *S. aureus*. After adhesion of T cells, *S. aureus* USA300 was added at an MOI1 and incubated at 37°C. After 3.5 hr, the samples were fixed with freshly prepared modified Karnovsky fixative (4% wt/vol PFA, 2.5% vol/vol glutaraldehyde in 0.1 M sodium cacodylate buffer, pH 7.4) for 1 hr at RT. After washing each three times for 15 min with 0.1 M sodium cacodylate buffer (pH 7.4), the cells were post-fixed with 2% (wt/vol) osmium tetroxide for 1 hr at RT. Subsequently, the samples were dehydrated in ascending ethanol concentrations (30%, 50%, 70%, 90%, and 100%) for 15 min each. Next, the samples were critical-point dried using liquid $CO_2$ and sputter coated with gold (thickness approx. 2 nm) using a CCU-010 sputter coater (safematic GmbH, Zizers, Switzerland). Finally, the specimens were investigated with a field emission SEM LEO-1530 Gemini (Carl Zeiss NTS GmbH, Oberkochen, Germany).

## Statistical analysis and scheme design

Data are presented as mean ± SD, or as median with interquartile range for violin and box plots, with up to four levels of statistical significance indicated. p-Values were calculated using Kruskal-Wallis test. Individual replicates are represented as single data points. Statistical analysis was carried out using GraphPad Prism v9.0.

## Acknowledgements

Breeding pairs of *Terc*[ko/ko] mice were kindly provided by Lenhard Rudolph (Leibniz Institute on Aging, Jena, Germany). We thank Sylvia Hänßgen, Yvonne Ozegowski, and Lea Herrmann for their excellent technical assistance. This work is supported by the BMBF, funding program Photonics Research Germany ('LPI-BT1-FSU', FKZ 13N15466; 'LPI-BT2-IPHT', FKZ 13N15704) and is integrated into the Leibniz Center for Photonics in Infection Research (LPI). The LPI initiated by Leibniz-IPHT, Leibniz-HKI, UKJ, and FSU Jena is part of the BMBF national roadmap for research infrastructures. We also want to thank the BMBF for the funding for the 'ADA' (13GW0456A). In addition, this work was supported by funding from the Foundation 'Else Kröner-Fresenius-Stiftung' within the Else Kröner Graduate School for Medical Students 'Jena School for Ageing Medicine (JSAM)'. This research was also supported by the Deutsche Forschungsgemeinschaft (DFG, German Research Foundation) under Germany's Excellence Strategy-EXC 2051 (Project ID No. 390713860).

## Additional information

### Funding

| Funder | Grant reference number | Author |
| --- | --- | --- |
| Bundesministerium für Bildung und Forschung | 13N15466 | Bettina Löffler |
| Bundesministerium für Bildung und Forschung | 13N15704 | Bettina Löffler |

| Funder | Grant reference number | Author |
|---|---|---|
| Bundesministerium für Bildung und Forschung | 13GW0456A | Stefanie Deinhardt-Emmer |
| Deutsche Forschungsgemeinschaft | 390713860 | Bettina Löffler |

The funders had no role in study design, data collection and interpretation, or the decision to submit the work for publication.

### Author contributions

Yasmina Reisser, Visualization, Methodology, Writing – original draft, Writing – review and editing; Franziska Hornung, Antje Häder, Thurid Lauf, Sandor Nietzsche, Visualization, Methodology, Writing – review and editing; Bettina Löffler, Resources, Supervision, Funding acquisition, Visualization, Writing – review and editing; Stefanie Deinhardt-Emmer, Conceptualization, Resources, Supervision, Funding acquisition, Visualization, Writing – original draft, Writing – review and editing

### Author ORCIDs

Yasmina Reisser ⓘ https://orcid.org/0009-0007-9757-5036
Stefanie Deinhardt-Emmer ⓘ https://orcid.org/0000-0003-4495-4052

### Ethics

Mouse in vivo experiments were approved by the Office for Consumer Protection of Thuringia (Permit Number: UKJ-19-028 and UKJ-22-023).

Reviewer #1 (Public review): https://doi.org/10.7554/eLife.100433.3.sa1
Reviewer #2 (Public review): https://doi.org/10.7554/eLife.100433.3.sa2
Author response https://doi.org/10.7554/eLife.100433.3.sa3

---

## Additional files

### Supplementary files

- MDAR checklist

- Supplementary file 1. Overview of RNA concentrations measured with Nanodrop and integrity values measured with the Agilent Bioanalyzer.

- Supplementary file 2. Data quality summary of each sample sequenced.

- Supplementary file 3. DEGs used to construct the individual heatmaps.

- Supplementary file 4. Exemplary results of genotyping of $Terc^{ko/ko}$ mice used for breeding.

- Source data 1. Zip folder containing the original gel image.

- Source data 2. Zip folder containing a PDF file of the original gel with all bands and samples labeled.

### Data availability

The data supporting this study's findings were made available as supplementary and source data files. The DEG lists generated via RNA sequencing were made available at Mendeley Data.

The following dataset was generated:

| Author(s) | Year | Dataset title | Dataset URL | Database and Identifier |
|---|---|---|---|---|
| Yasmina R, Stefanie DE | 2024 | Telomerase RNA component knockout exacerbates *S. aureus* pneumonia by extensive inflammation and dysfunction of T cells | https://data.mendeley.com/datasets/bvpzvn5sbg/1 | Mendeley Data, 10.17632/bvpzvn5sbg.1 |

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
