## [Editor Report · eLife Assessment]

In this manuscript, the authors sought to elucidate mechanistic intricacies of inflammatory responses, with emphasis on T cell dysfunction, to *S. aureus*-induced pneumonia in the context of aging process using Terc deficient mice. Conceptually, the study is very interesting with a set of **useful** findings. Although some experimental approaches are appropriate, the work as shown in the revised manuscript remains significantly underpowered and the absence of rigorous controls make this study **incomplete** in support of its claims.

---

## [Referee Report · Reviewer #1 (Public review)]

Summary:

This work sets out to elucidate mechanistic intricacies in inflammatory responses in pneumonia in the context of aging process (Terc deficiency - telomerase functionality).

Strengths:

Very interesting, conceptually speaking, approach that is by all means worth pursuing. An overall proper approach to the posited aim.

Weaknesses:

The work is heavily underpowered and may have statistical deficits. This precludes at its current state drawing unequivocal conclusions.

I remain at my initial position regarding the weaknesses.

---

## [Referee Report · Reviewer #2 (Public review)]

Summary

The authors demonstrate heightened susceptibility of Terc-KO mice to *S. aureus*-induced pneumonia, perform gene expression analysis from the infected lungs, find an elevated inflammatory (NLRP3) signature in some Terc-KO but not control mice, and some reduction in T cell signatures. Based on that, they conclude that dysregulated inflammation and T cell dysfunction play a major role in these phenomena.

The strengths of the work did not change, and include a problem not previously addressed (the role of Terc component of the telomerase complex) in certain aspects of resistance to bacterial infection and innate (and maybe adaptive) immune function.

The weaknesses of this revised version still outweigh the strengths, because the authors did not substantially or experimentally answer the main criticism points, and have rather tried to argue away that which cannot be argued away. In summary, the most germane conclusions of this study remain plagued by flaws in experimental design, by lack of rigorous controls and by incomplete and inadequate approaches to testing of immune function.

I will devote the rest of the comments to the revised manuscript and its success or lack thereof in responding to prior criticisms. Prior criticisms are again listed below in italics, to provide context for the attempts of the investigators to respond.

(1) Reviewer 1 has justifiably criticized the exceptionally low power of the study, with 5 control and 3 experimental animals. The responding author has replied that the animal welfare laws preclude them from doing more experiments. That is unfortunate, and I sympathize with the authors. Nonetheless, in the absence of robust corroboration the rigor of the study remains severely compromised and the work is reduced to what I have pointed above - a preliminary and inconclusive study that is in need of deeper and more serious mechanistic investigation.

(2) Terc-KO mice are a genomic knockout model, and therefore the authors need to carefully consider the impact of this KO on a wide range of tissues. This, however, is not the case. There are no attempts to perform cell transfers, use irradiation chimera or crosses that would be informative.

In response to this criticism, the authors have quoted a whole bunch of papers characterizing different aspects of biology of these same mice. The most important paper in that regard would be the one by Matthe et al. on CD4 cells from these same mice. That study was limited and simply diagnosed in situ the changes in T cell pool, but did not decipher whether and to what extent such defects are cell-intrinsic or a byproduct of similarly altered microenvironments. Most importantly, none of that answers the original critique question of which cell types are truly the culprits in the Terc deletion phenotype presented here. As I indicated, one has to perform cell transfers, bone marrow irradiation chimera, additional genetic crosses and combinations thereof to substantiate whether the defects are ascribable to the lung tissue itself, the infiltrating myeloid cells, including macrophages, the T cells or a combination thereof. The authors provided none of this.

(3) Throughout the manuscript the authors invoke the role of telomere shortening in aging, and according to them their Terc-KO mice should be one potential model for aging. Yet the authors consistently describe major differences between young Terc-KO and naturally aging old mice, with no discussion of the implications. This further confuses the biological significance of this work as presented.

(4) Related to #2, group design for comparisons lacks a clear rationale. The authors stipulate that Terc-KO will mimic natural aging, but in fact, the only significant differences seen between groups in susceptibility to *S. aureus* are, contrary to the authors' expectation, between young Terc-KO and naturally old mice (Fig. 1A and B, no difference between young Terc-KO and young wt); or there are no significant differences at all between groups (Fig. 1, C, D,). I have also raised the issue of non-physiological nature of a germline Terc-KO, that does not mimic any known physiological or pathological state.

The authors provided a non-response to this criticism. They argue in their response under (2) of their rebuttal that they included old mice as controls not for aging, because their experimental Terc-deletion mice were G3 and do not exhibit as much of a progeroid phenotype as G5 or G6 mice. But they still say in the revised formulation that these mice were infected "to explore the potential link to a fully developed aging phenotype". They just never conclude that no such link is substantiated by the vast majority of their data. Moreover, they come back to state in their response (4) that because the literature reported ".... reduction of Terc and Tert in tissues of old mice and rats. Therefore, as a potential immunomodulatory factor reduced Terc expression could be connected to age-related pathologies." So either they have used old mice here to compare aging phenotypes, and found that Terc-KO mice diverge massively from aging phenotypes, in which case they have to state so, or they are not using them as age comparators (in which case I am not sure what their purpose is).

(5) (originally part of criticism #4) I have criticized inadequate group design is when the authors begin dividing their Terc-KO groups by clinical score into animals with or without "systemic infection" (the condition where a bacterium spreads uncontrollably across the many organs and via blood, which should be properly called sepsis), and then compare this sepsis group to other groups (Suppl Fig. 1G; Fig. 2; lines 374-376 and 389-391). .... Most importantly, methodologically it is highly inappropriate to compare one mouse with sepsis to another one without. If Terc-KO mice with sepsis are a comparator group, then their controls have to be wild type mice with sepsis, who are dealing with the same high bacterial load across the body and are presumably forced to deploy the same set of immune defenses.

The authors responded by making me aware of the 2016 JAMA definition of sepsis that invokes "a life-threatening organ dysfunction caused by a dysregulated host response to infection". I appreciate the correction, and note that in a human setting and globally, such a definition may make sense. The authors stated that bacteremia and not sepsis should be used as a criterion. I agree, and per my original criticism, believe it will be appropriate to compare bacteremic wt and KO mice.

(6) I am shortening my prior critique to make it more to the point that was not addressed: The authors conclude that disregulated inflammation and T cell dysfunction play a major role in *S. aureus* susceptibility. This may or may not be an important observation, because many KO mice are abnormal for a variety of reasons, and until such reasons are mechanistically dissected, the physiological importance of the observation will remain unclear. ....., the authors truly did not examine the key basic features of their model, including the features of basic and induced inflammatory and immune response. This analysis could be done either using model antigens in adjuvants, defined innate immune stimuli (e.g. TLR, RLR or NLR agonsists), or microbial challenge. The only data provided along these lines are the baseline frequencies of total T cells in the spleen of the three groups of mice examined (not statistically significant, Fig. 4B). We do not know if the composition of naïve to memory T cell subsets may have been different, and more importantly, we have no data to evaluate whether recruitment of the immune response (including T cells) to the lung upon microbial challenge is similar or different. So, what are the numbers and percentages of T cells and alveolar macrophages in the lung following *S. aureus* challenge and are they even comparable or are there issues in mobilizing the T cell response to the site of infection ? If, for example, Terc-KO mice do not mobilize enough T cells to the lung during infection, that would explain paucity in many T cell -associated genes in their transcriptomic set that they authors report. That in turn may not mean dysfunction of T cells but potentially a whole different set of defects in coordinating the response in Terc-KO mice.

The authors did not respond to this criticism other than to provide more frequencies of different subsets. The key here are the NUMBERS of cells present at the peak of challenge, or better yet the kinetics of cell accumulation (again numbers), as well as transfer experiments to establish where the defect actually lies (mobilization, activation, proliferation, etc.).

(7) Related to that, immunological analysis is also inadequate. First, the authors pull signatures from the total lung tissue, which is both imprecise and potentially skewed by differences not in gene expression but in types of cells present and/or their abundance, a feature known to be affected by aging and perhaps by Terc deficiency during infection. Second, to draw any conclusions about immune responses, the authors would have to track antigen-specific T cells, which is possible for a wide range of microbial pathogens using peptide-MHC multimers. This would allow highly precise analysis of phenomena the authors are trying to conclude about. Moreover, it would allow them to confirm their gene expression data in populations of physiological interest.

The authors agreed that this would be of interest but did nothing to provide it. They provided a sentence in the discussion stating that this (as well as many other experiments needed to interpret the results) would be of interest.

(8) Overall, the authors begun to address the role of Terc in bacterial susceptibility, but to what extent that specifically involves inflammation and macrophages, T cell immunity or aging remains unclear at the present.

My conclusion from the prior review remains unchanged in the face of the revision that did not answer most of the previous criticism. The study as it stands is inconclusive and highly preliminary, with lack of clearly defined mechanistic underpinnings.

---

## [Author Response]

The following is the authors’ response to the original reviews.

**Public Reviews:**

**Reviewer #1 (Public Review):**
Summary:This work sets out to elucidate mechanistic intricacies in inflammatory responses in pneumonia in the context of the aging process (Terc deficiency - telomerase functionality).Strengths:Very interesting, conceptually speaking, approach that is by all means worth pursuing. An overall proper approach to the posited aim.

We want to thank the reviewer for taking the time to review our manuscript and for providing positive feedback regarding our research question.

Weaknesses:The work is heavily underpowered and may have statistical deficits. This precludes it in its current state from drawing unequivocal conclusions.

Thank you for this essential and valuable comment. We fully accept that the small sample size of the Tercko/ko mice is a major limitation of our study and transparently discuss this in our manuscript. However, due to Animal Welfare regulations, only a reduced number of mice were approved because of the strong burden of disease. Consequently, only three non-infected and five infected mice were available to us. This reduced number of mice presents a clear limitation to our study. However, due to ethical considerations related to animal welfare and sustainability, as well as compliance with German animal welfare regulations, it is not possible to obtain additional Tercko/ko mice to increase the dataset.

The animal studies are an important aspect of our study; however, our hypothesis was also investigated at multiple levels, including in an in vitro co-culture model (Figure 5), to ensure comprehensive analysis. Thus, we clearly demonstrated that *S. aureus* pneumonia in Tercko/ko mice leads to a more severe phenotype, orchestrated by the dysregulation of both innate and adaptive immune response.

**Reviewer #2 (Public Review):**
Summary:The authors demonstrate heightened susceptibility of Terc-KO mice to *S. aureus*-induced pneumonia, perform gene expression analysis from the infected lungs, find an elevated inflammatory (NLRP3) signature in some Terc-KO but not control mice, and some reduction in T cell signatures. Based on that, They conclude that disregulated inflammation and T-cell dysfunction play a major role in these phenomena.Strengths:The strengths of the work include a problem not previously addressed (the role of the Terc component of the telomerase complex) in certain aspects of resistance to bacterial infection and innate (and maybe adaptive) immune function.

We would like to thank the reviewer for the positive feedback regarding our aim to investigate the impact of *Terc* deletion on the pulmonary immune response to *S. aureus*.

Weaknesses:The weaknesses outweigh the strengths, dominantly because conclusions are plagued by flaws in experimental design, by lack of rigorous controls, and by incomplete and inadequate approaches to testing immune function. These weaknesses are as follows(1) Terc-KO mice are a genomic knockout model, and therefore the authors need to carefully consider the impact of this KO on a wide range of tissues. This, however, is not the case. There are no attempts to perform cell transfers or use irradiation chimera or crosses that would be informative.

We thank the reviewer for bringing up this important point. The aim of our study, however; was to investigate the impact of *Terc* deletion in the lung and on the response to bacterial pneumonia, rather than to provide a comprehensive characterization of the Tercko/ko model itself. This characterization of different tissues and cell types has already been conducted by previous studies. For instance, studies that characterize the general phenotype of the model (Herrera et al., 1999; Lee et al., 1998; Rudolph et al., 1999) but also investigations that shed light on the impact of Terc deletion on specific cell types such as microglia (Khan et al., 2015) or T cells (Matthe et al., 2022). The impact of *Terc* deletion on T cells is also discussed in our manuscript in lines 89 to 105. Furthermore, a section about the general phenotype of the *Terc* deletion model is included in the introduction in lines 126 to 138. Thus we discussed the relevant literature regarding Tercko/ko mice in our manuscript and attempted to provide a more in-depth characterization of the lung by investigating the inflammatory response to infection as well as changes in the gene expression (Figure 2-4).

(2) Throughout the manuscript the authors invoke the role of telomere shortening in aging, and according to them, their Terc-KO mice should be one potential model for aging. Yet the authors consistently describe major differences between young Terc-KO and naturally aging old mice, with no discussion of the implications. This further confuses the biological significance of this work as presented.

Thank you for mentioning this relevant point. We want to apologize for the confusion regarding this matter. While Tercko/ko mice are a well-established model for premature aging, these effects become more apparent with increasing generations (G) and thus, G5 and 6 mice are the most affected by *Terc* deletion (Lee et al., 1998; Wong et al., 2008).

Thus, while Tercko/ko mice are a common model for premature aging, this accelerated aging phenotype is predominantly apparent in later-generation Tercko/ko (G5 and 6) or aged Tercko/ko mice (Lee et al., 1998; Wong et al., 2008). Since the aim of this study was to analyze the impact of *Terc* deletion on the lung and its immune response to bacterial infections instead of the impact of telomere shortening and telomerase dysfunction, young G3 Tercko/ko mice (8 weeks) were used in this study. This is also mentioned in the lines 131-134. In this study, Tercko/ko mice were used not as a model of aging, but rather as a model specifically for *Terc* deletion. The old WT mice function as a control cohort to observe possible common but also deviating effects between aging and *Terc* deletion. In our sequencing data, we observe that uninfected young WT mice are very similar to uninfected Tercko/ko mice. Other studies have also reported this lack of major differences between uninfected WT and Tercko/ko mice in the G3 knockout mice (Kang et al., 2018). Conversely, uninfected young WT and Tercko/ko mice exhibited great differences, for instance, regarding the numbers of differentially expressed genes (Supplemental Figure 1H). Thus, differences between naturally aged mice and young G3 Tercko/ko mice are not surprising. To clarify this aspect we reconstructed the paragraph discussing the Tercko/ko mice (lines 126-134). Additionally we added a paragraph explaining the purpose of the naturally aged mice to the lines 134 to 138:

“As control cohort age-matched young WT mice were utilized. To investigate whether *Terc* deletion, beyond critical telomere shortening, impacts the pulmonary immune response, we used young Tercko/ko mice. Additionally, naturally aged mice (2 years old) were infected to explore the potential link to a fully developed aging phenotype.”

(3) Related to #2, group design for comparisons lacks a clear rationale. The authors stipulate that TercKO will mimic natural aging, but in fact, the only significant differences seen between groups in susceptibility to *S. aureus* are, contrary to the authors' expectation, between young Terc-KO and naturally old mice (Figures 1A and B, no difference between young Terc-KO and young wt); or there are no significant differences at all between groups (Figures 1, C, D,).

We thank the reviewer for this essential comment. As mentioned above the Tercko/ko mice in this study are not selected to model natural aging. To model telomerase dysfunction and accelerated aging selection of later generation or aged Tercko/ko mice would have been more suitable.

The lack of statistical significance in some figures is likely due to the heterogeneity of disease phenotype of *S. aureus* infection in mice, which is a limitation of our study that we discuss in our discussion section in lines 576-582. The phenotype of *S. aureus* infection can vary greatly within a mouse population, highlighting the limitations of mice as a model for *S. aureus* infections. To account for this heterogeneity we divided the infected Tercko/ko mice cohort into different degrees of severity based on the clinical score and the presence of bacteria in organs other than the lung (mice with systemic infection).

Despite the heterogeneity especially within the Tercko/ko mice cohort the differences between the knockout and young as well as old WT mice were striking. Including the fatal infections, 80% of the Tercko/ko mice had a severe course of disease, while none of the WT mice displayed a severe course (Figure 1A, B and Supplemental Figure 1A, B). This hints towards a clear role of *Terc* in the response to *S. aureus* infection in mice. Thus while in some figures the differences are not significant, strong trends towards a more severe phenotype of *S. aureus* infection in the Tercko/ko mice regarding bacterial load, score and inflammatory response could be observed in our study.

Another example of inadequate group design is when the authors begin dividing their Terc-KO groups by clinical score into animals with or without "systemic infection" (the condition where a bacterium spreads uncontrollably across the many organs and via blood, which should be properly called sepsis), and then compare this sepsis group to other groups (Supplementary Figures 1G; Figure 2; lines 374-376 and 389391). This gives them significant differences in several figures, but because they did not clearly indicate where they applied this stratification in the figure legends, the data are somewhat confusing. Most importantly, methodologically it is highly inappropriate to compare one mouse with sepsis to another one without. If Terc-KO mice with sepsis are a comparator group, then their controls have to be wild-type mice with sepsis, who are dealing with the same high bacterial load across the body and are presumably forced to deploy the same set of immune defenses.

We sincerely appreciate the significant time and effort you have invested in reviewing our manuscript. However, with all due respect, we must point out that the definition of sepsis you have referenced is considered outdated. According to the Third International Consensus Definitions for Sepsis and Septic Shock (Sepsis-3), sepsis is defined as "a life-threatening organ dysfunction caused by a dysregulated host response to infection" (Marvin Singer, 2016, JAMA). Given this fundamental misunderstanding of our findings, we find the comment regarding the inadequacy of our groups to be both dismissive and lacking in scientific merit. We would like to emphasize that the group size used in our study is consistent with accepted standards in infection research. We strongly reject any insinuations of inadequacy that have been repeatedly mentioned throughout the review.

In order to provide a nuanced investigation of disease severity in Tercko/ko mice, we added the term “systemic infection” to the figures whenever the mice were divided into groups of mice with and without systemic infection. This is the case for Figure 2A and Supplemental Figure 1C-E. The division into mice with and without systemic infection is also mentioned in the figure legend of Figure 2A in lines 932 to 935 and for Supplemental Figure 1 in lines 1052-1053. We agree that Supplemental Figure 1G is somewhat confusing as the mice with systemic infection are highlighted in this graph but not included as a separate group within our sequencing analysis. We added a sentence to the figure legend clarifying this (lines 1042-1044):

“Nevertheless, the infected Tercko/ko mice were considered one group for the expression analysis and not split into separate groups for the subsequent analysis.”

Additionally, we revised the section regarding this grouping in different degrees of severity in our Material and Methods section to clarify that this division was only performed for specific analysis (line 191):

“…for the indicated analysis.”

Furthermore, the mice which were classified as systemically infected mice were not septic mice, as mentioned above. Those mice were classified by us as systemically infected based on their clinical score and the presence of bacteria in other organs than the lung as stated in the lines 188-191 and 377-381. Bacteremia is a symptom of very severe cases of hospital-acquired pneumonia with a very high mortality (De la Calle et al., 2016).

Therefore, the systemically infected mice or rather mice with bacteremia display an especially severe pneumonia phenotype, which is distinct from sepsis. The presence of this symptom in our Tercko/ko mice further highlights the clinical relevance of our study. This aspect was added to the manuscript in the lines 568-570.

“The detection of bacteria in extra pulmonary organs is of particular interest, as bacteremia is a symptom of severe pneumonia and is associated with high mortality (De la Calle et al., 2016).”

(4) The authors conclude that disregulated inflammation and T-cell dysfunction play a major role in *S. aureus* susceptibility. This may or may not be an important observation, because many KO mice are abnormal for a variety of reasons, and until such reasons are mechanistically dissected, the physiological importance of the observation will remain unclear.Two points are important here. First, there is no natural counterpart to a Terc-KO, which is a complete loss of a key non-enzymatic component of the telomerase complex starting in utero.Second, the authors truly did not examine the key basic features of their model, including the features of basic and induced inflammatory and immune responses. This analysis could be done either using model antigens in adjuvants, defined innate immune stimuli (e.g. TLR, RLR, or NLR agonists), or microbial challenge. The only data provided along these lines are the baseline frequencies of total T cells in the spleen of the three groups of mice examined (not statistically significant, Figure 4B). We do not know if the composition of naïve to memory T cell subsets may have been different, and more importantly, we have no data to evaluate whether recruitment of the immune response (including T cells) to the lung upon microbial challenge is similar or different. So, what are the numbers and percentages of T cells and alveolar macrophages in the lung following *S. aureus* challenge and are they even comparable or are there issues in mobilizing the T cell response to the site of infection? If, for example, Terc-KO mice do not mobilize enough T cells to the lung during infection, that would explain the paucity in many T-cellassociated genes in their transcriptomic set that the authors report. That in turn may not mean dysfunction of T cells but potentially a whole different set of defects in coordinating the response in Terc-KO mice.

We thank the reviewer for highlighting these important aspects. Regarding the first point, indeed there is no naturally occurring deletion of *Terc* in humans. However, studies reported reduced expression of *Terc* and *Tert* in the tissues of aged mice and rats (Tarry-Adkins et al., 2021; Zhang et al., 2018). *Terc* itself has been found to have several important immunomodulatory functions such as the activation of the NFκB or PI3-kinase pathway (Liu et al., 2019; Wu et al., 2022). As those aforementioned pathways are relevant for the immune response to *S. aureus* infections, the authors were interested in exploring the impact of *Terc* deletion on the pulmonary immune response. The potential immunomodulatory functions of *Terc* are discussed in lines 106-121. To further clarify our rationale we added a sentence to the introduction in lines 121-125.

“Interestingly, downregulation of *Terc* and *Tert* expression in tissues of aged mice and rats has been found (Tarry-Adkins, Aiken, Dearden, Fernandez-Twinn, & Ozanne, 2021; Zhang et al., 2018). Therefore, as a potential immunomodulatory factor reduced *Terc* expression could be connected to agerelated pathologies.”

Regarding the second point, as we focused on the effect of *Terc* deletion in the lung and its role in *S. aureus* infection, we investigated inflammatory and immune response parameters relevant to this setting. For instance, inflammation parameters in the lungs of all three mice cohorts were measured to investigate differences in the inflammatory response in the non-infected and infected mice (Figure 2A). Those measurements showed no baseline difference in key inflammatory parameters between young WT and Tercko/ko mice, which is consistent with previous findings (Kang et al., 2018). The inflammatory response to infection with *S. aureus* in the Tercko/ko mice cohort differed significantly from the other cohorts (Figure 2A), hinting towards a dysregulated inflammatory response due to *Terc* deletion. Furthermore, we investigated general immune cell frequencies such as dendritic cells, macrophages, and B cells in the spleen of all three mice cohorts to gather a baseline understanding of the general immune cell populations. In our manuscript only total T cell frequencies were included due to its relevance for our data regarding T cells (Figure 4B). This data could show that there was no difference of total amount of T cells in the spleen of all three mice cohorts. For a more detailed insight into our analysis we added the frequencies of the other immune cell populations analyzed in the spleen as a Supplemental Figure 3B-F. Additionally, a figure legend for the graphs was added to lines 1075-1094.

Therefore, while we did not analyze baseline frequencies of specific populations of T cells, we analyzed and characterized the inflammatory and immune response of our model in a way relevant to our research question.

The differences observed in T cell marker and TCR gene expression was also partly present between the uninfected and infected Tercko/ko mice such as the complete absence of *CD247* expression in infected Tercko/ko, which is however expressed in uninfected mice of this cohort (Figure 4A, C and D). Thus, this effect cannot be solely attributed to an inadequate mobilization of T cells to the lung after infectious challenge. However, we agree that a more detailed insight into recruited immune cells to the lung or frequencies of different T cell populations could contribute to a better understanding of the proposed mechanism and would be an interesting experiment to conduct in further studies. We accept this as a limitation of our study and included it in our discussion section in lines 719-723:

“As total CD4+ T cells were analyzed in this study, it would be useful to investigate specific T cell populations such as memory and effector T cells to elucidate the potential mechanism leading to T cell dysfunctionality in further detail. Additionally, analysis of differences in immune cell recruitment to the lungs between young WT and Tercko/ko mice would be relevant.”

(5) Related to that, immunological analysis is also inadequate. First, the authors pull signatures from the total lung tissue, which is both imprecise and potentially skewed by differences, not in gene expression but in types of cells present and/or their abundance, a feature known to be affected by aging and perhaps by Terc deficiency during infection. Second, to draw any conclusions about immune responses, the authors would have to track antigen-specific T cells, which is possible for a wide range of microbial pathogens using peptide-MHC multimers. This would allow highly precise analysis of phenomena the authors are trying to conclude about. Moreover, it would allow them to confirm their gene expression data in populations of physiological interest

We thank the reviewer for highlighting this important and relevant point. In our study, we aimed to investigate the role of *Terc* expression in modulating inflammation and the immune response to *S. aureus* infection in the lung. To address this, we examined the overall impact of age, genotype, and infection on lung inflammation and gene expression. Therefore, sequencing of total lung tissue was essential for addressing the research question posed. Our findings demonstrate that Tercko/ko mice exhibit a more severe phenotype following *S. aureus* infection, characterized by an increased bacterial load and heightened lung inflammation (Figures 1 and 2). Furthermore, our data suggest that *Terc* plays a role in regulating inflammation through activation of the NLRP3 inflammasome, along with the dysregulation of several T cell marker genes (Figures 2, 4, and 5). However, this study lacks a detailed analysis of distinct T cell populations, including antigen-specific T cells, as noted earlier. Investigating these aspects in future studies would be valuable to validate and expand upon our findings. We have incorporated these suggestions into the discussion section (lines 719-723)

“As total CD4+ T cells were analyzed in this study, it would be useful to investigate specific T cell populations such as memory and effector T cells to elucidate the potential mechanism leading to T cell dysfunctionality in further detail. Additionally, analysis of differences in immune cell recruitment to the lungs between young WT and Tercko/ko mice would be relevant.”

Nevertheless, our study provides first evidence of a potential connection between T cell functionality and *Terc* expression.

Third, the authors co-incubate AM and T cells with *S. aureus*. There is no information here about the phenotype of T cells used. Were they naïve, and how many S. aureus-specific T cells did they contain? Or were they a mix of different cell types, which we know will change with aging (fewer naïve and many more memory cells of different flavors), and maybe even with a Terc-KO? Naïve T cells do not interact with AM; only effector and memory cells would be able to do so, once they have been primed by contact with dendritic cells bringing antigen into the lymphoid tissues, so it is unclear what the authors are modeling here. Mature primed effector T cells would go to the lung and would interact with AM, but it is almost certain that the authors did not generate these cells for their experiment (or at least nothing like that was described in the methods or the text).

Thank you for bringing up this important question. For the co-cultivation experiment of T cells and alveolar macrophages, total CD4+ T cells of both young WT and Tercko/ko were used. We did not select for a specific population of T cells. Our sequencing data indicated the complete downregulation of *CD247* expression, which is an important part of the T cell receptor, in the lungs of infected Tercko/ko mice (Figure 4A, C and D). Given that this factor is downregulated under chronic inflammatory conditions, we investigated the impact of the inflammatory response in alveolar macrophages on the expression of various T cell-derived cytokines, as well as *CD247* expression (Figure 5D, E) (Dexiu et al., 2022). This aspect is also highlighted in the discussion in lines 622-636. Therefore, a co-cultivation model of T cells and alveolar macrophages was established and confronted with heat-killed *S. aureus* to elicit an inflammatory response of the macrophages. To emphasize this purpose, we have revised our statement about the model setup in lines 516-518 of the manuscript:

“An overactive inflammatory response could be a potential explanation for the dysregulated TCR signaling.”

The authors hope this will clarify the intent behind the model setup.

(6) Overall, the authors began to address the role of Terc in bacterial susceptibility, but to what extent that specifically involves inflammation and macrophages, T cell immunity, or aging remains unclear at present.

We thank the reviewer for the helpful and relevant comments. The authors accept the limitations of the presented study such as the reduced number of Tercko/ko mice and the limitations of murine models for *S. aureus* infection itself and discuss those in the discussion section in the lines 558-560; 576-582; 688-690 and 719-725. However, we hope that our responses have provided sufficient evidence to convince the reviewer that our data supports a clear role for *Terc* expression in regulating the immune response to bacterial infections, particularly with respect to inflammation and its potential connection to T cell functionality.

**Recommendations for the authors:**

**Reviewer #1 (Recommendations For The Authors):**
The good element first:I read this paper with genuine interest and applaud the authors for investigating the posited question. I consider it by all means scientifically relevant in the context of physiological/pathophysiological aging and reaction to a disease (here pneumonia). The Terc deletion model looks very appropriate for the question and the methodology is very advanced/in-depth. The data flow/selection of endpoints and assays is very logical to me. Moreover, I like the breakdown of pneumonia into varying levels of severity.

We thank the reviewer for their time and effort taken to revise our manuscript. Additionally, we are grateful to receive your positive feedback regarding our study design and research question.

The weaknesses:(1) I cannot help but notice that the study is heavily underpowered. As such, it is inadmissible. The key reason is that it is the first of its kind and seminal findings must be strongly propped by the evidence. It is apparent to me that the data scatter presented in the figures tends to be abnormally distributed (e.g. obvious bimodal distribution in some groups). Therefore, the presented comparisons (even if stat. sign) can be heavily misleading in terms of: (i) the true magnitude of the observed effects and (ii) possible type 2 error in some cases of p value >0.05. Solution: repeat the study to ensure reasonable power/reliability. This will also make it stronger as it will immediately demonstrate its reproducibility (or lack of it).

Thank you for bringing up this extremely relevant point. We acknowledge the issue of the small sample size of Tercko/ko mice as a major limitation of our study. This limitation is also included in our discussion section in the lines 558-560. Thus we fully agree with this limitation and transparently discuss this in our manuscript. However, due to the strict German animal welfare regulations it is not possible to obtain more Tercko/ko mice, as mentioned above. Furthermore, since fatal infections occurred in the Tercko/ko mice cohort we had a reduced number of mice available.

However, the differences between the Tercko/ko and WT mice were striking. Including the fatal infections 80% of the Tercko/ko mice had a severe course of disease, while none of the WT mice displayed a severe course. This hints towards a clear role of *Terc* in the response to *S. aureus* infection in mice.

(2) In the stat analysis section of M&Ms, the authors feature only 1 sentence. This cannot be. A detailed stats workup needs to be included there. This is very much related to the above weakness; e.g. it is impossible to test for normality (to choose an appropriate post-hoc test) with n=3. Back to square one: study underpowered.

We thank the reviewer for highlighting this important aspect. We carefully revised the method section in lines 357-360 to include all relevant information:

“Data are presented as mean ± SD, or as median with interquartile range for violin and box plots, with up to four levels of statistical significance indicated. P-values were calculated using Kruskal-Wallis test. Individual replicates are represented as single data points.”

(3) Pneumonia severity. While I noted that as a strength, I also note it as weakness here. It looks to me like the authors stopped halfway with this. I totally support testing a biological effect(s) such as the one investigated here across a spectrum of a given disease severity. The authors mention that they had various severity phenotypes produced in their model but this is not visible in the data figs. I strongly suggest including that as well; i.e., to study the posited question in the severe and mild pneumonia phenotype. This is a very smart path and previous preclinical research clearly demonstrated that this severe/mild distinction is very relevant in the context of the observed responses (their presence/absence, longevity, dynamics, etc). I realize this is challenging, thus, I would probably use this approach in the Terc k/o model as sort of a calibrator to see whether the exacerbation observed in the current setup (severe?) will be also present in a mild pneumonia phenotype. *S. aureus* can be effectively titrated to produce pneumonia of varying severity.

We thank the reviewer for bringing up this relevant point.

In our study, we could observe heterogeneity within the infected Tercko/ko cohort. Therefore as pointed out by the reviewer we assigned different degrees of severity to those groups based on clinical scores, the fatal outcome of the disease (fatal subgroup), and the presence of bacteria in organs other than the lungs (systemic infection subgroup) as stated in our materials and methods part in the lines 188-191 (Supplemental Figure 1A and B). Moreover, we highlighted this difference in a number of our figures. For example, when categorizing the mice into groups with and without systemic infection, we noticed that the mice with systemic infection demonstrated a higher bacterial load, significant body weight loss, and increased lung weight (see Supplemental Figure 1C-E). Interestingly, the two mice with systemic infection clustered separately from the other mice, indicating that the mice with systemic infection are transcriptomically distinct from the other mice cohorts (Supplemental Figure 1G). Additionally, the inflammatory response was exclusively elevated in the lungs of mice with systemic infection (Figure 2C). Thus, we included this distinction in several figures and attempted to study the differences between those subgroups but also their similarities. For instance, we could observe that some changes in the transcriptome were present in all three infected Tercko/ko mice such as the complete absence of *CD247* expression at 24 hpi (Figure 4D). This distinction therefore provided a more detailed insight into the underlying mechanisms of disease severity in Tercko/ko mice and is lacking in other studies. We agree with the reviewer, that a study investigating mild and severe pneumonia phenotypes would be clinically relevant. However, as noted above, due to ethical considerations related to animal welfare and sustainability, as well as compliance with German animal welfare regulations, it is not possible to obtain additional Tercko/ko mice to carry out the proposed experiment.

(4) Please read ARRIVE guidelines and note the relevant info in M&Ms as ARRIVE guidelines point out.

Thank you for emphasizing this crucial aspect. We revised our materials and methods section according to the ARRIVE guidelines (lines 179-206).

“Tercko/ko mice aged 8 weeks, were used for infection studies (n = 8; non-infected = 3; infected = 5). Female young WT (age 8 weeks) and old WT (age 24 months) C57Bl/6 mice (both n = 10; non-infected = 5; infected = 5) were purchased from Janvier Labs (Le Genest-Saint-Isle, France). All infected mouse cohorts were compared to their respective non-infected controls, as well as to the infected groups from other cohorts. Additionally, comparisons were made between the non-infected cohorts across all groups.

All mice were anesthetized with 2% isoflurane before intranasal infection with *S. aureus* USA300 (1x108 CFU/20µl) per mouse. After 24 hours, the mice were weighed and scored as previously described (Hornung et al., 2023). Infected Tercko/ko mice were grouped into different degrees of severity based on their clinical score, fatal outcome of the disease (fatal) and the presence of bacteria in organs other than the lung (systemic infection) for the indicated analysis. Mice with fatal infections were excluded from subsequent analyses, with only their final scores being reported. The mice were sacrificed via injection of an overdose of xylazine/ketamine and bleeding of axillary artery after 24 hpi. BAL was collected by instillation and subsequent retrieval of PBS into the lungs. Serum and organs were collected. Bacterial load in the BAL, kidney and liver was determined by plating of serially diluted sample as described above. For this organs were previously homogenized in the appropriate volume of PBS. Gene expression was analyzed in the right superior lung lobe. Lobes were therefore homogenized in the appropriate amount of TriZol LS reagent (Thermo Fisher Scientific, Waltham, MA, US) prior to RNA extraction. The left lung lobe was embedded into Tissue Tek O.C.T. (science services, Munich, Germany) and stored at 80°C until further processing for histological analysis. Cytokine measurements were performed using the right inferior lung lobe. Lobes were previously homogenized in the appropriate volume of PBS. Remaining organs were stored at -80°C until further usage. Mouse studies were conducted without the use of randomization or blinding.“

(5) There are also some other descriptive deficits but they are of a much smaller caliber so I do not list them.

We thank the reviewer for their valuable and insightful suggestions for improving our manuscript. We hope that our responses and the corresponding revisions address these suggestions satisfactorily.

Concluding: the investigative idea is great/interesting and the methodological flow is adequate but the low power makes this study of low reliability in its current form. I strongly urge the authors to walk the extra mile with this work to make it comprehensive and reliable. Best of luck!
**Reviewer #2 (Recommendations For The Authors):**
(1) Many legends are uninformative and do not contain critical information about the experiments. For example, Figure 2A with cytokine measurements (in lung homogenates?) is likely showing data from an ELISA or Luminex test, but there is no mention of that in the legend. It stands next to Figure 2B, which is a gene expression map, again, likely from the lung (prepared how, normalized how, etc?) lacking even the most basic information. Further, Figure 2D has no information on the meaning/effect size of gene ratios on the x-axis. Figures 3 and 4 are presumably the subsets of their transcriptome data set (whole lung, harvested on d ?? post-infection), but that is just a guess on my part. Even in the main text, the timing and the controls for the transcriptomic study are not stated (ln. 398 and onwards). The authors really need to revise the figure legends and provide all the details that an average reader would need to be able to interpret the data.

We thank the reviewer for bringing up this important point. The figure legends of all figures including supplemental figures were revised to ensure they include all relevant data necessary for accurate interpretation of the graphs. Additionally, we clarified the sequenced samples in lines 427-429:

“We performed mRNA sequencing of the murine lung tissue of infected and non-infected mice at 24 hpi to elucidate potential differentially expressed genes that contribute to the more severe illness of Tercko/ko mice.”

(2) Telomere shortening affects differentially different cells and its role in aging is nuanced - different in mesenchymal cells with no telomerase induction, in non-replicating cells, and in hematopoietic cells that can readily induce telomerase. The authors should be mindful of that in setting up their introduction and discussion.

Thank you for mentioning this essential aspect. We revised our introduction and discussion to reflect the nuanced role of telomerase shortening in different tissues (lines 83-92 and 690-695):

“Telomerase activity is restricted to specific tissues and cell types, largely dependent on the expression of *Tert*. While *Tert* is highly expressed in stem cells, progenitor cells, and germline cells, its expression is minimal in most differentiated cells (Chakravarti, LaBella, & DePinho, 2021). Consequently, the impact of telomerase dysfunction on tissues varies according to their self-renewal rate. (Chakravarti et al., 2021). One important aspect of telomere dysfunction is the impact of telomere shortening on the immune system as well as the hematopoietic system. Tissues or organ systems that are highly replicative, such as the skin or the hematopoietic system, are affected first by telomere shortening (Chakravarti et al., 2021).”

“It is important to note that telomere shortening has a significant impact on the immune system. Although young Tercko/ko mice were used in this study, telomere shortening is still likely to be a contributing factor. Therefore, further experiments investigating the role of T cell senescence in this model should therefore be conducted.”

(3) Syntax and formulations need to be improved and made more scientifically precise in several spots. Specifically, in 62-63, the authors say that the aged immune system "is also discussed to be more irritable", please change to reflect the common notion that the reaction to infection is dysregulated; in many cases inflammation itself is initially blunted, misdirected, and of different type (e.g. for viruses, the key IFN-I responses are not increased but decreased). In lines 114-117, presumably, the two sentences were supposed to be connected by a comma, although some editing for clarity is probably needed regardless. Line 252, please change "unspecific" to "non-specific". Line 264, please capitalize German.

We thank the reviewer for bringing these important points to our attention. We revised our introduction regarding the aged immune response in lines 61-69:

“Age-related dysregulation of the immune response is also characterized by inflammaging, defined as the presence of elevated levels of pro-inflammatory cytokines in the absence of an obvious inflammatory trigger (Franceschi et al., 2000; Mogilenko, Shchukina, & Artyomov, 2022). Additionally, immune cells, such as macrophages, exhibit an activated state that alters their response to infection (Canan et al., 2014). In contrast, the immune response of macrophages to infectious challenges has been shown to be initially impaired in aged mice (Boe, Boule, & Kovacs, 2017). Thus aging is a relevant factor impacting the pulmonary immune response.”

Sentences were edited to provide more clarity in lines 131-134:

“Although G3 Tercko/ko mice with shortened telomeres were used in this study, they were infected at a young age (8 weeks). This approach allowed for the investigation of *Terc* deletion effects rather than telomere dysfunction.”

“Unspecific was changed to “non-specific” in line 282 and “German” was capitalized in line 293 and 558.

We appreciate and thank you for your time spent processing this manuscript and look forward to your response.

References

De la Calle, C., Morata, L., Cobos-Trigueros, N., Martinez, J. A., Cardozo, C., Mensa, J., & Soriano, A. (2016). *Staphylococcus aureus* bacteremic pneumonia. *European Journal of Clinical Microbiology & Infectious Diseases*, *35*(3), 497-502. https://doi.org/10.1007/s10096-015-2566-8

Dexiu, C., Xianying, L., Yingchun, H., & Jiafu, L. (2022). Advances in CD247. *Scand J Immunol*, *96*(1), e13170. https://doi.org/10.1111/sji.13170

Herrera, E., Samper, E., Martín-Caballero, J., Flores, J. M., Lee, H. W., & Blasco, M. A. (1999). Disease

states associated with telomerase deficiency appear earlier in mice with short telomeres. *Embo j*, *18*(11), 2950-2960. https://doi.org/10.1093/emboj/18.11.2950

Hornung, F., Schulz, L., Köse-Vogel, N., Häder, A., Grießhammer, J., Wittschieber, D., Autsch, A., Ehrhardt, C., Mall, G., Löffler, B., & Deinhardt-Emmer, S. (2023). Thoracic adipose tissue contributes to severe virus infection of the lung. *International Journal of Obesity*, *47*(11), 10881099. https://doi.org/10.1038/s41366-023-01362-w

Kang, Y., Zhang, H., Zhao, Y., Wang, Y., Wang, W., He, Y., Zhang, W., Zhang, W., Zhu, X., Zhou, Y., Zhang, L., Ju, Z., & Shi, L. (2018). Telomere Dysfunction Disturbs Macrophage Mitochondrial Metabolism and the NLRP3 Inflammasome through the PGC-1α/TNFAIP3 Axis. *Cell Reports*, *22*(13), 3493-3506. https://doi.org/10.1016/j.celrep.2018.02.071

Khan, A. M., Babcock, A. A., Saeed, H., Myhre, C. L., Kassem, M., & Finsen, B. (2015). Telomere dysfunction reduces microglial numbers without fully inducing an aging phenotype. *Neurobiology of Aging*, *36*(6), 2164-2175. https://doi.org/10.1016/j.neurobiolaging.2015.03.008

Lee, H.-W., Blasco, M. A., Gottlieb, G. J., Horner, J. W., Greider, C. W., & DePinho, R. A. (1998). Essential role of mouse telomerase in highly proliferative organs. *Nature*, *392*(6676), 569-574. https://doi.org/10.1038/33345

Liu, H., Yang, Y., Ge, Y., Liu, J., & Zhao, Y. (2019). TERC promotes cellular inflammatory response independent of telomerase. *Nucleic Acids Research*, *47*(15), 8084-8095. https://doi.org/10.1093/nar/gkz584

Matthe, D. M., Thoma, O. M., Sperka, T., Neurath, M. F., & Waldner, M. J. (2022). Telomerase deficiency reflects age-associated changes in CD4+ T cells. *Immun Ageing*, *19*(1), 16. https://doi.org/10.1186/s12979-022-00273-0

Rudolph, K. L., Chang, S., Lee, H. W., Blasco, M., Gottlieb, G. J., Greider, C., & DePinho, R. A. (1999). Longevity, stress response, and cancer in aging telomerase-deficient mice. *Cell*, *96*(5), 701-712. https://doi.org/10.1016/s0092-8674(00)80580-2

Tarry-Adkins, J. L., Aiken, C. E., Dearden, L., Fernandez-Twinn, D. S., & Ozanne, S. (2021). Exploring Telomere Dynamics in Aging Male Rat Tissues: Can Tissue-Specific Differences Contribute to Age-Associated Pathologies? *Gerontology*, *67*(2), 233-242. https://doi.org/10.1159/000511608

Wong, L. S. M., Oeseburg, H., de Boer, R. A., van Gilst, W. H., van Veldhuisen, D. J., & van der Harst, P. (2008). Telomere biology in cardiovascular disease: the TERC−/− mouse as a model for heart failure and ageing. *Cardiovascular Research*, *81*(2), 244-252. https://doi.org/10.1093/cvr/cvn337

Wu, S., Ge, Y., Lin, K., Liu, Q., Zhou, H., Hu, Q., Zhao, Y., He, W., & Ju, Z. (2022). Telomerase RNA TERC and the PI3K-AKT pathway form a positive feedback loop to regulate cell proliferation independent of telomerase activity. *Nucleic Acids Res*, *50*(7), 3764-3776. https://doi.org/10.1093/nar/gkac179

Zhang, M. W., Zhao, P., Yung, W. H., Sheng, Y., Ke, Y., & Qian, Z. M. (2018). Tissue iron is negatively correlated with TERC or TERT mRNA expression: A heterochronic parabiosis study in mice. *Aging (Albany NY)*, *10*(12), 3834-3850. https://doi.org/10.18632/aging.101676